# Optical heterostructure in a two-dimensional organic crystal

Kan Liao[1,2,6], Junran Zhang [1,6], Xiang-Long Yu [3,6], Wenheng Xu[1], Zhongjing Xia[1], Dawei Zhou[4], Zilong Mao[1], Yan Lv [1], Yijun Ming[1], Chao Liu[4], Ming Sheng[4], Kun Liu[1], Zhen Zhang[2], Chongqin Zhu [5], Xiaoyong Wang [2], Chao Zhu [4] ✉, Zhongfu An [1] ✉ & Lin Wang [1] ✉

Optical heterostructures, which feature spatially heterogeneous optical properties at the subwavelength scale, represent a key frontier for next-generation integrated photonics and optoelectronics. While typically realized by joining dissimilar materials, achieving such heterogeneity in single-component systems has remained a fundamental challenge. Here, we report an intrinsic optical heterostructure in a uniform organic nanosheet, manifesting as strongly enhanced fluorescence in the inner zone compared to the outer zone. We demonstrate that this emission heterogeneity stems from a spatially localized solid-state transition in the central top layer, which transforms the initial single crystal into an out-of-plane twin structure and significantly enhances the radiative recombination efficiency. This transition is driven by the competitive interplay between molecule-substrate and intermolecular interactions, as corroborated by multiscale structural, optical, and theoretical analyses. Our findings not only establish a platform for realizing optical heterostructures in organic materials but also open avenues for exploring structural-dynamics-governed photonic phenomena, offering broad implications for future materials design and micro-optical applications.

A heterostructure is composed of heterogeneous materials with different properties, which exhibit superior functionalities compared to homogeneous materials[1]. Similarly, an optical heterostructure is defined as a system consisting of two or more units with different optical properties, which spatially exhibit rich and unique optical properties with respect to individual material[2,3]. Owing to their considerable application potential, these structures have garnered significant interest in materials science. They offer a promising platform for high-resolution optical coding, data encryption, anti-counterfeiting, optical multiplexing, and multi-wavelength lasing[2–4]. Furthermore, the unique combination of spatially confined luminescence and tunable efficiency positions them as compelling candidates for developing next-generation optoelectronic components, including on-chip light sources, microscale lasers, and integrated photonic circuits[3,5,6]. Typically, optical heterostructures are achieved by lateral connection of materials with distinct chemical components or lattice structures. However, the mismatch between these materials often

[1]State Key Laboratory of Flexible Electronics, School of Flexible Electronics (Future Technologies) & Institute of Advanced Materials, School of Physical and Mathematical Sciences, Nanjing Tech University, Nanjing, China. [2]National Laboratory of Solid State Microstructures, School of Physics, and Collaborative Innovation Center of Advanced Microstructures, Nanjing University, Nanjing, China. [3]School of Science, Sun Yat-sen University, Shenzhen, China. [4]SEU-FEI Nano-Pico Center, Key Laboratory of MEMS of Ministry of Education, School of Integrated Circuits, Southeast University, Nanjing, China. [5]College of Chemistry, Key Laboratory of Theoretical & Computational Photochemistry of Ministry of Education, Beijing Normal University, Beijing, China. [6]These authors contributed equally: Kan Liao, Junran Zhang, Xiang-Long Yu. ✉e-mail: zhuchao@seu.edu.cn; iamzfan@njtech.edu.cn; iamlwang@njtech.edu.cn

necessitates intricate design and artificial control of assembling processes[2,3,7].

Comparatively, optical heterostructures in single-component system offer distinct advantages, including simple chemical fabrication strategies, minimal lattice mismatches, and avoidable defects. Recent advancements in inorganic single-component systems have demonstrated the realization of optical heterostructure through the induction of crystalline phase transitions, grain boundary formations, defect engineering, and other structural differences[5,8–10]. Within these layers, atoms are interconnected by robust covalent bonds, as depicted at the top of Fig. 1a, forming solid periodic lattices. These microscopic structural differences can easily disrupt the strict lattice periodicity and influence the energy band details, ultimately affecting the macroscopic optical performance along in-plane direction[2,6,7,11].

In contrast, the observation of optical heterostructure in organic single-component systems is much more challenging due to the fundamental differences between organic and inorganic materials. Organic crystals hold weak van der Waals forces along both the in-plane and interlayer directions, as illustrated at the bottom of Fig. 1a, compared to the strong covalent bonds that unite atoms within

inorganic layers. Consequently, organic materials exhibit significant in-plane structural flexibility and fault tolerance, making it difficult to effectively regulate macroscopic properties via microscopic structural differences[12,13]. Additionally, the lack of experimental strategies to control molecular packing orientation further complicates the precise creation of structural difference in organic materials. Therefore, the realization of optical heterostructures within organic materials has primarily relied on approaches such as cavity effects or multi-component systems[3,14–18].

In this work, we demonstrate the intrinsic optical heterostructure achieved in a single organic-component nanosheet. The inner and outer zones of such a nanosheet exhibit distinct photoluminescence (PL) behaviors, despite having no visual difference in composition, lattice structure, and thickness. The optical heterostructure is attributed to a twin structure phase transition occurring in the top inner part of the nanosheet, driven by the synergistic and competitive cooperation between intermolecular forces and molecule-substrate interactions. Our findings highlight the crucial role of interfacial interaction, beyond intermolecular interactions, in designing promising structures and functionalities for organic materials.

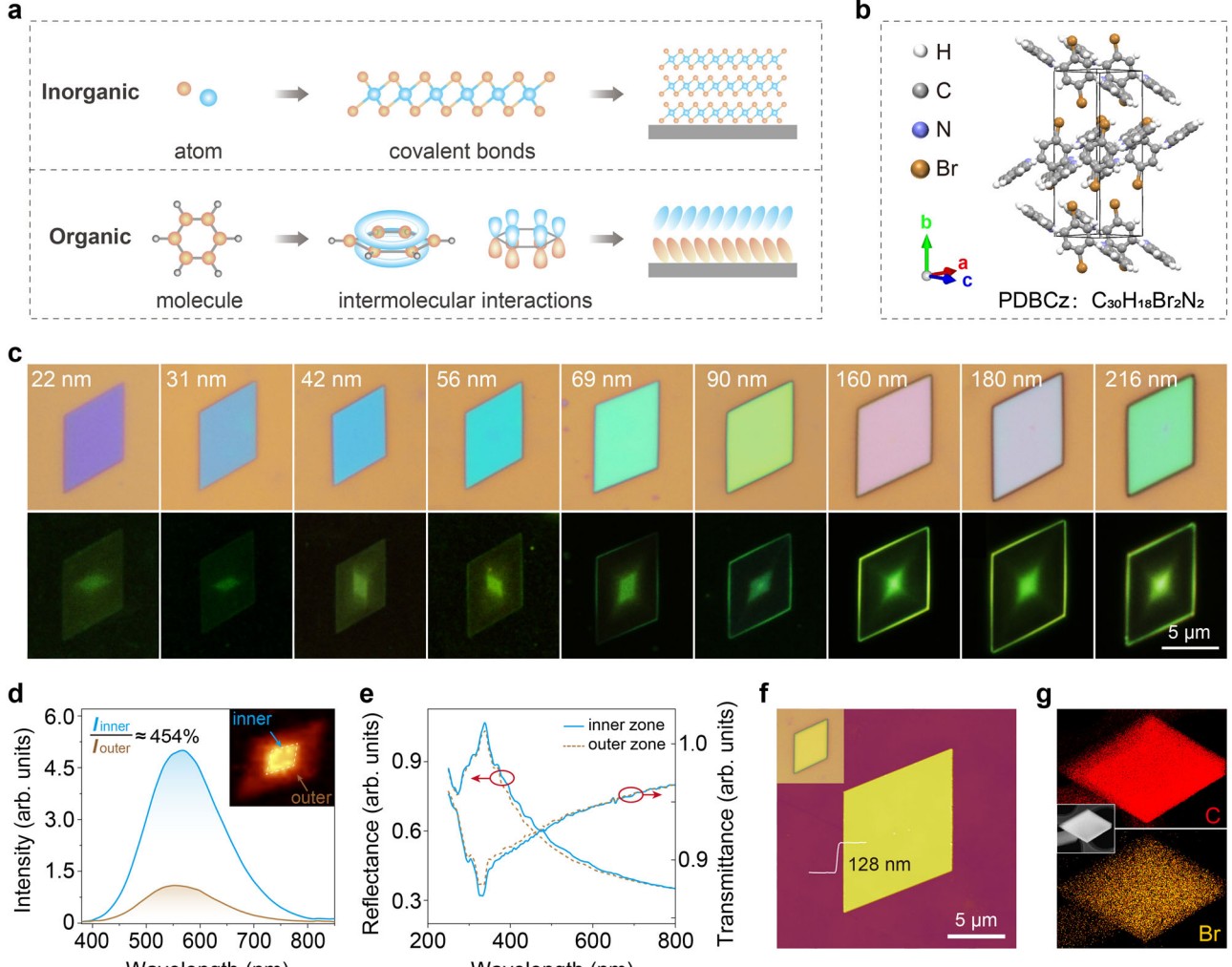

**Fig. 1 | Optical heterostructure observed in a single organic-component system. a** Schematic diagram of atom/molecule packing in inorganic/organic crystals. **b** Unit cell structure of PDBCz. **c** The optical (top) and fluorescence (bottom) images of PDBCz nanosheets with different thicknesses. **d** Photoluminescence (PL) spectra of the inner and the outer zone in a typical 128 nm-thick nanosheet, recorded at room temperature under excitation of 405 nm. Inset shows the PL mapping image, where the colors represent PL intensity (pseudo-color) rather than the actual emission color. **e** Transmittance and reflectance spectra collected from the inner and outer zones. **f** Atomic force microscopy image of the nanosheet. Inset is the optical image. **g** Transmission electron microscopy coupled with energy-dispersive X-ray spectroscopy (TEM-EDS) images of the nanosheet with uniform distribution of C (red) and Br (yellow) elements throughout the entire nanosheet. Inset shows the TEM image.

## Results and discussion

### Optical heterostructure observed in a single organic-component system

The optical heterostructure was serendipitously discovered in nanosheets of single organic component 9,9'-(2,5-dibromo-1,4-phenylene)bis[9H-carbazole] ($C_{30}H_{18}Br_2N_2$, hereafter referred as PDBCz). These nanosheets were fabricated using the physical vapor deposition (PVD) method, which prevents any unintended chemical reactions and enable the nanosheets with single component. X-ray diffraction (XRD) analysis of nanosheets is consist with that of single-crystals (Supplementary Fig. 1), which confirms the crystal phase and demonstrates the space group of $P2_1/c$[19]. Figure 1b and Supplementary Fig. 2 depict the unit cell structure. Optical microscopy images presented in the upper panel of Fig. 1c demonstrate that regardless of their varying thicknesses, the nanosheets exhibit a uniformly smooth surface and a distinctive parallelogram morphology.

A prominent feature of these nanosheets is their intrinsic optical heterostructure, characterized by a consistently brighter inner zone as revealed by fluorescence imaging (Fig. 1c bottom) and PL mapping (Fig. 1d inset). Spectral analysis of a representative PDBCz nanosheet shows the PL intensity in the inner zone ($I_{inner}$) is ~454% stronger than the outer zone ($I_{outer}$), despite sharing an identical emission maximum at ~552 nm (Fig. 1d). Critically, the near-overlapping transmittance and reflectance spectra of both zones (Fig. 1e) indicate comparable optical absorption and photon excitation rates[20], collectively pointing to a superior radiative recombination efficiency in the inner zone as the sole origin of the enhanced emission (Supplementary Note I). The regular geometric shape of this high-efficiency region starkly contrasts with the disorderly zones in inorganic analogs due to randomly formed grain boundaries, defects, or phases[5,8-10]. It is also noteworthy that the outer edge can exhibit brighter emission due to optical confinement effects like waveguide modes[17,21], which is distinct from the optical heterostructure reported here.

We systematically investigated the origin of the optical heterostructure. Our initial hypothesis of optical microcavity effects was definitively ruled out, as the enhanced emission in the inner zone persists even when the nanosheet's regular shape is disrupted (Supplementary Fig. 3) Optical image and atomic force microscopy (AFM) image (Fig. 1f) revealed no visible differences in the surface morphologies between the inner and outer zones. Notably, transmission electron microscopy coupled with energy-dispersive X-ray spectroscopy (TEM-EDS) (Fig. 1g) shows that the distribution of elements (such as carbon and bromine) is uniform throughout the entire nanosheet. Furthermore, the crystal structures of the inner and outer zones are highly consistent, as evidenced by the matching selected area electron diffraction (SAED) patterns obtained from different zones (Fig. 2a and Supplementary Figs. 4, 5). Therefore, the optical heterostructure phenomenon cannot be attributed to differences in surface morphology, elemental distribution, or crystal structure between the inner and outer zones.

### Twin structure of PDBCz nanosheets

Strikingly, crystallographic analysis revealed a twin structure along the b-axis in PDBCz, which is seldom observed in organic crystals. This is definitively demonstrated by the selected area electron diffraction patterns (Fig. 2a and Supplementary Figs. 4, 5), as the electron diffraction patterns from both the inner and outer zones consistently displayed two sets of spots. These patterns match perfectly with the simulated twinned $m_1 + m_2$ structure with mirror symmetry (Fig. 2b), in clear contrast to the single set of spots expected for a single crystal (Fig. 2c). Typically, the formation of twin structures is the result of cooperative effects of multiple interactions, which is challenging to achieve in organic materials, especially in bulk form. When organic crystals become molecularly thin, however, the molecule-substrate interaction becomes comparable and competitive with intermolecular interactions[22,23]. Therefore, the most likely reason for the formation of the twin structure in PDBCz nanosheets is the cooperative effects of intermolecular and molecule-substrate interactions.

We designed a series of control experiments to investigate the interactions between the PDBCz nanosheets and substrates. Solution-processed methods are known to yield free-standing materials with ignorable molecule-substrate interaction. As shown in Fig. 2d, the PDBCz nanosheets prepared by solution method exhibit only one set of diffraction pattern, with no twin structure or optical heterostructure phenomenon. Similarly, PDBCz nanosheets grown on other types of non-siliceous substrates (such as BN, mica, and sapphire substrates) using PVD methods also lack twin structure and do not exhibit optical heterostructure phenomenon as shown in Fig. 2e and Supplementary Fig. 6. In contrast, PDBCz nanosheets grown on various siliceous substrates (such as $SiO_2/Si$, Si, and quartz substrates) all exhibit twin structure and the optical heterostructure phenomenon (Supplementary Fig. 7). To further probe the role of bromine, we examined structurally similar molecules without Br atoms, such as 1,4-di-(-H-carbazol-9-yl)benzene (referred as PDCz)[24] and 1.3-bis(N-carbazoly)benzene (referred as mCP)[25]. When grown on silicon substrates via the same PVD method, these Br-free nanosheets show no optical heterostructure (Supplementary Fig. 8), highlighting the critical role of Br. Given the potential for covalent Si-Br bonding[26], we calculated the adsorption and intermolecular interaction energies for these systems (Supplementary Note II and Supplementary Table 1). The results reveal a distinct covalent Br-Si interaction between PDBCz and $SiO_2$ (Fig. 2f and Supplementary Fig. 9), which is absent for PDBCz on BN, as well as for Br-free PDCz on either substrate (Fig. 2f, g). Energetically, the PDBCz-$SiO_2$ system exhibits a markedly lower adsorption energy (−0.764 eV) than its intermolecular interaction energy (−0.723 eV), whereas the PDCz-$SiO_2$ system shows the opposite trend (Fig. 2h). This contrast confirms that the PDBCz-$SiO_2$ interface features strong binding—primarily mediated by Si-Br bonding—which surpasses both the material's own intermolecular cohesion and the adsorption of Br-free analogs.

### Island growth of PDBCz nanosheets

To gain insights into the formation and evolution of the twin structure, we conducted in-situ and real-time investigations on the dynamic growth process of PDBCz nanosheets, as shown in the bottom of Fig. 3a. From the optical images, we observed that the crystal nuclei initially appear as separated islands with circular or elliptical shapes. As the growth time increases, these nuclear islands gradually grow to fractal and compact islands, and then form into nanosheets. This growth process follows an island growth mode (top of Fig. 3a)[27,28], which differs from the layer-by-layer growth mode commonly observed for organic crystals[22,29,30]. Guided by island growth, we calculated the surface energies and relative growth rates of the major crystal facets to unravel the growth kinetics (Supplementary Note III). The (020) surface exhibits a distinctly lower free energy than other facets (Fig. 3c), leading to its slowest growth rate and thereby defining the crystal morphology as the largest exposed face—a finding consistent with our experimental results (Fig. 3a, bottom). Overall, the observed island growth mode with a substrate-parallel orientation represents the most energetically favored configuration.

The temporal evolution of the structure and optical properties provides direct insight into the underlying mechanism. Initially, neither the optical heterostructure nor the twin structure is observed (Fig. 3b). They emerge simultaneously once the nanosheet develops a regular parallelogram shape and are conserved during subsequent growth, establishing a direct correlation between the structural transition and the emergent optical phenomenon.

Based on this, we propose a coherent growth model (Fig. 3e). In the initial stage (I), strong molecule-substrate interactions (Fig. 2f, g, h) enforce an identical $m_1$ lattice in both layers, which is dominant for

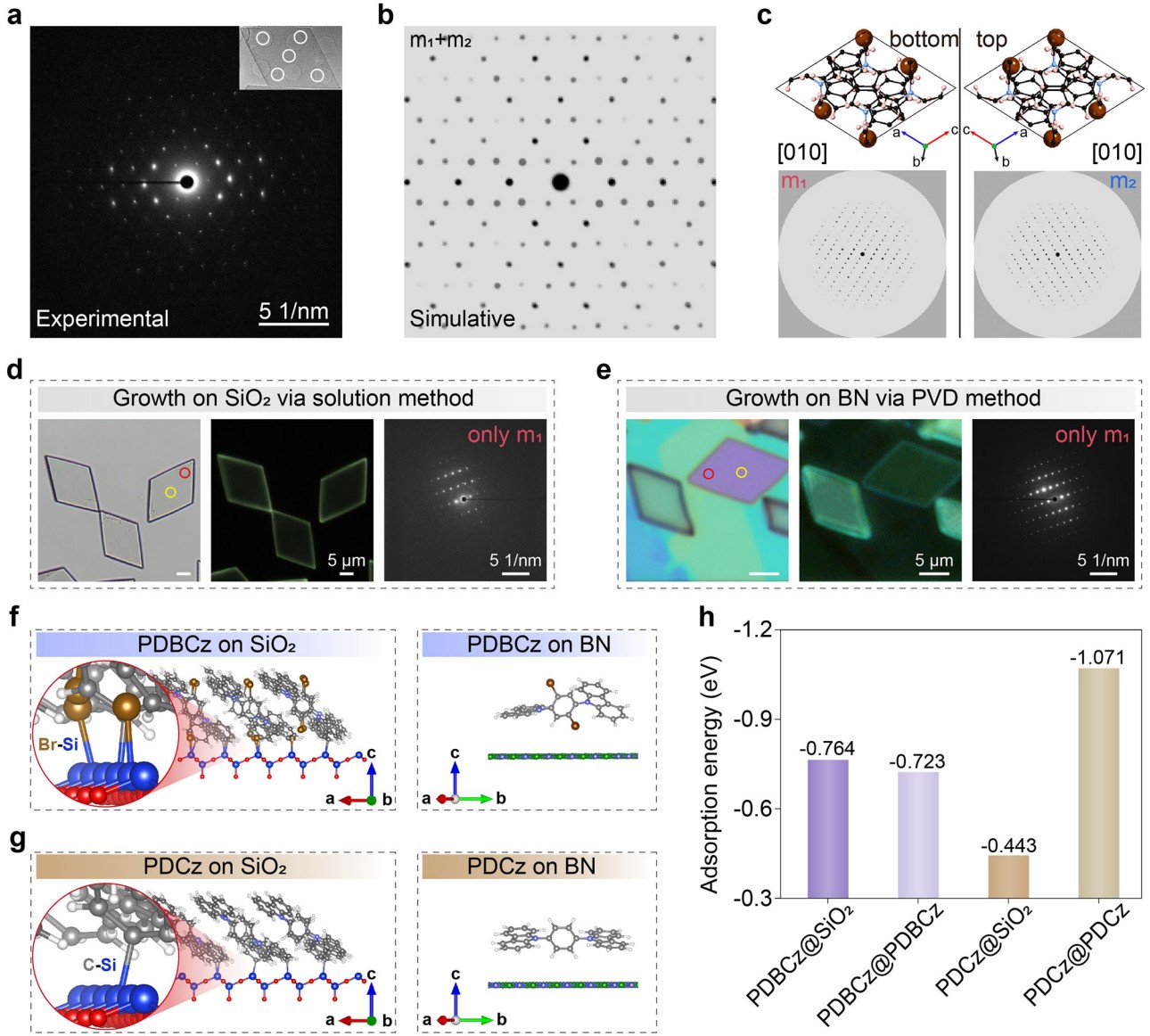

**Fig. 2 | Twin structure of PDBCz nanosheets. a** TEM-SAED pattern of a PDBCz nanosheet. This displays superimposed diffraction patterns captured at various zones of the nanosheet denoted by white circles in the inset TEM image, all of which closely coincide with each other, indicating the uniform crystal structure over the entire nanosheet. **b** Simulative diffraction pattern of PDBCz with twin structure. **c** Two separated simulative diffraction patterns with mirror symmetry and the corresponding molecular structures. **d**, **e** Optical, fluorescence, and SAED images of PDBCz nanosheets prepared by solution method on SiO₂ substrate and by physical vapor deposition (PVD) method on BN substrate, respectively, in which the SAED images from different zones as marked by red and yellow circles are the same. **f**, **g** Theoretical simulations of PDBCz and PDCz molecules on SiO₂ and BN substrates, respectively. **h** Theoretically calculated adsorption energies of PDBCz-SiO₂ and PDCz-SiO₂ interfaces, compared with their intermolecular interaction energy.

small-sized nanomaterials[22,23]. As the nanosheet grows, intermolecular interactions become increasingly influential. Upon reaching a critical size (Stage II), this interplay triggers a localized solid-state transition exclusively in the top layer, which rotates into the $m_2$ orientation, forming a mirror-twin structure ($m_1 + m_2$) with the substrate-pinned bottom layer. This process is fundamentally enabled by strong specific interactions with the siliceous substrate. During subsequent lateral growth (Stage III), the system propagates this heterostructure, with the bottom and top layers perpetuating the $m_1$ and $m_2$ lattices, respectively, thereby preserving the twin configuration and the associated optical heterostructure.

To elucidate why the lateral growth of the top layer consistently maintains the $m_2$ orientation in the outer zone rather than reverting to $m_1$ after the structural transition, we conducted structural relaxation simulations (Fig. 3d and Supplementary Fig. 10). These calculations

demonstrate that when $m_1$ and $m_2$ molecules are positioned adjacently within the same $a$-$c$ plane, the system undergoes a spontaneous reorientation toward a uniform molecular arrangement. This confirms that mixed $m_1$-$m_2$ packing is energetically unfavorable in the lateral plane and is intrinsically eliminated during structural optimization. The result provides a theoretical basis for the observed structural stability, explaining how the coherent $m_2$ domain extends into the outer zone during continued growth while preserving the well-defined twin boundary.

## Mechanism of transition-induced optical heterostructure formation

Building on this structural model, we present direct experimental evidence linking the localized twin transition to the optical heterostructure. First, we employed Kelvin probe force microscopy (KPFM)

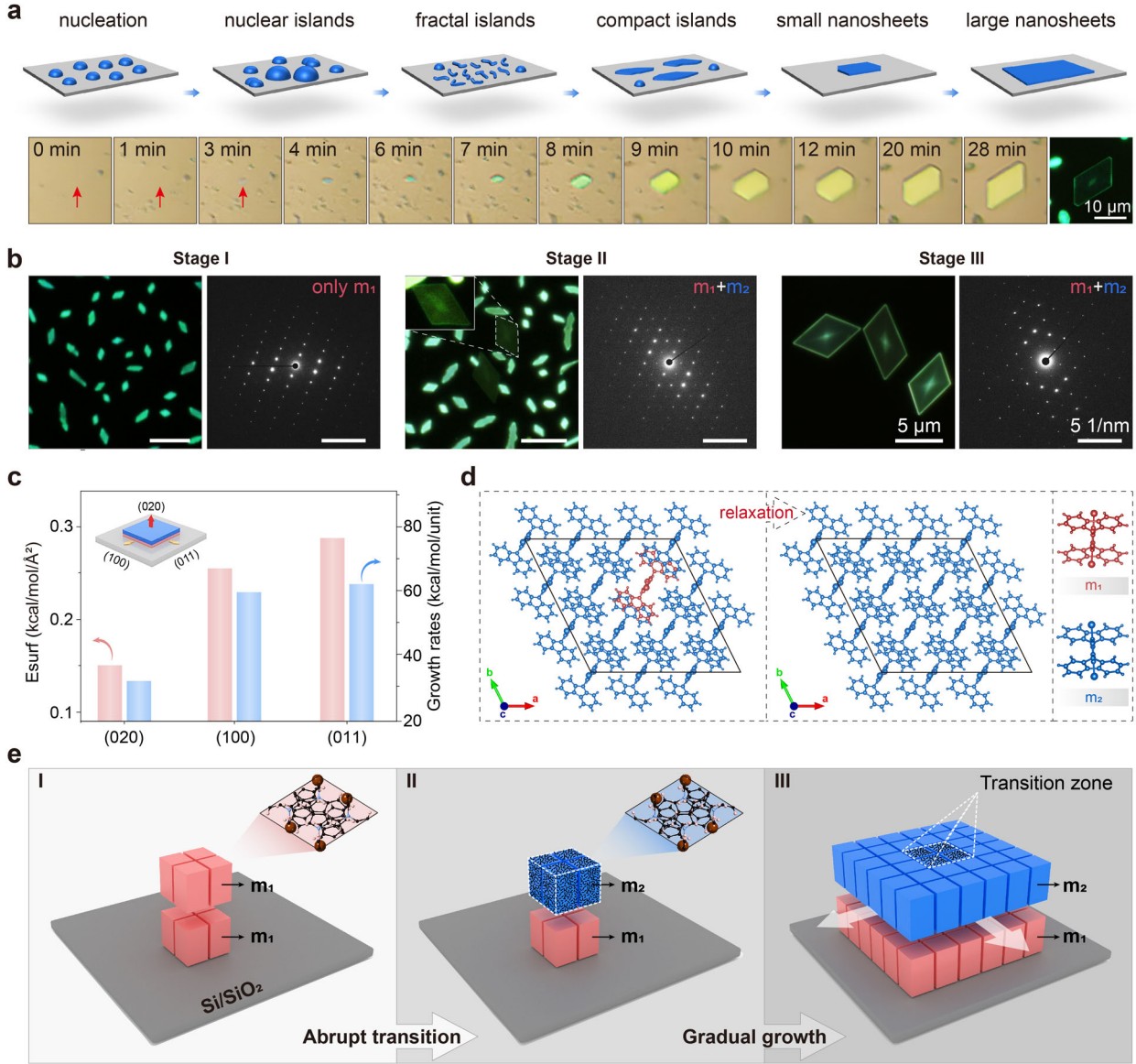

**Fig. 3 | Island growth of PDBCz nanosheets. a** Tops are schematic diagram of island growth mode. Bottoms are in-situ and real-time optical images of PDBCz nanosheets during growth process. **b** Fluorescence and SAED images of PDBCz nanosheets at three typical growth stages. **c** Theoretically calculated surface free energies and corresponding growth rates of the (100), (011), and (020) facets. Inset is the crystal growth model. **d** Molecular arrangements before and after structural relaxation. The inner blue region corresponds to the $m_1$ structure, which is enclosed by an outer layer of $m_2$ molecules. **e** Schematic diagram of the optical heterostructure mechanism of a PDBCz nanosheet with three important stages of growth process.

and electrostatic force microscopy (EFM) to probe the local electronic and structural properties. These techniques sensitively map variations in surface potential and electrostatic force gradients, which serve as indicators of crystallinity and structural heterogeneity[31,32]. On a thick, as-grown nanosheet, we observed a pronounced contrast in both surface potential and EFM phase between the inner and outer zones (Fig. 4a), consistent with the crystallographic difference induced by the structural transition. Second, we leveraged the inherent fragility of materials at structural transition regions[33–35]. Thermal annealing of PVD-grown nanosheets on siliceous substrates caused degradative damage exclusively within the inner zone (Fig. 4a), directly confirming that the structural transition is spatially confined to this region. In contrast, solution-grown nanosheets, which lack the transition, showed no such differential damage (Supplementary Fig. 11). Together, these results unequivocally correlate the confined structural transition with the emergence of the optical heterostructure.

To further validate that the structural transition is confined to the top layer of the inner zone, we investigated its dependence on nanosheet thickness. According to our model, the transition requires a sufficient vertical distance from the substrate and should therefore be absent in sufficiently thin nanosheets. This was confirmed by two complementary approaches. First, we progressively thinned thick nanosheets via plasma etching (Supplementary Note IV and Supplementary Fig. 12). As the thickness decreased, the luminescence contrast between the inner and outer zones diminished accordingly, and vanished completely at a critical thickness of ~20 nm (Supplementary Figs. 13–15). Concurrently, the contrast in KPFM and EFM images also disappeared (Supplementary Fig. 15). Second, as-grown thin nanosheets (≤ 20 nm) inherently exhibited no optical heterostructure nor any KPFM/EFM contrast (Fig. 4b). Critically, neither the artificially thinned nor the as-grown thin nanosheets (including solution-processed ones) showed any damage after thermal annealing (Fig. 4b and Supplementary Fig. 16), in stark contrast to the localized damage observed in

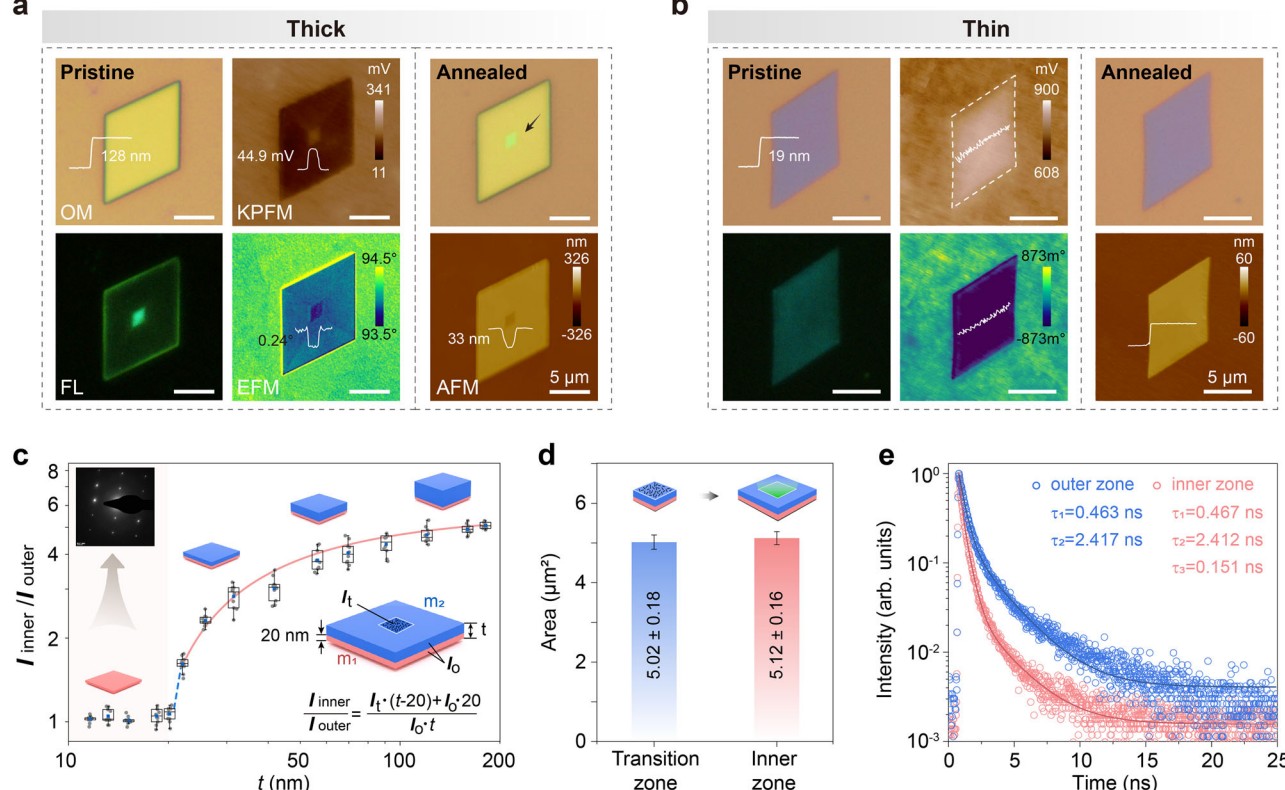

**Fig. 4 | Mechanism of transition-induced optical heterostructure formation.**
**a, b** Optical microscope (OM), fluorescence (FL), Kelvin probe force microscopy
(KPFM) and electrostatic force microscopy (EFM) images of a pristine thick PDBCz
nanosheet (~128 nm) and a pristine thin nanosheet (~19 nm), respectively. After
annealing, the OM and atomic force microscope (AFM) images reveal significant
structural degradation localized in the central region of the thick nanosheet,
whereas the thin nanosheet remains intact. **c** PL intensity ratio between the inner
and the outer zone ($I_{inner}/I_{outer}$) of nanosheets with different thicknesses. The
experimental results (solid dots, presented as box plots) are fitted to the proposed
scenario using the formula shown in the right inset (pink lines). The left inset shows
the SAED pattern of an as-grown thin nanosheet (~20 nm). **d** Statistical analysis of
the areas of the transition and PL-enhanced inner zones. Data are presented as
mean values with error bars representing the standard error (SE) calculated for
each individual data point ($n = 60$). **e** Time-resolved PL (TRPL) spectra of the inner
and outer zones, revealing a shorter lifetime and an additional fast radiative
recombination pathway in the inner zone.

thick nanosheets. Furthermore, selected area electron diffraction
confirmed that these thin nanosheets exhibit only a single-crystal
pattern without any twin feature (Fig. 4c, inset). Collectively, these
results demonstrate that a minimum thickness is required for the
structural transition to occur, confirming its localization to the top
layer and directly linking the transition to the emergence of the optical
heterostructure.

The thickness-dependent behavior of the structural transition was
quantitatively verified by analyzing the PL intensity ratio ($I_{inner}/I_{outer}$)
across nanosheets of different thicknesses (Supplementary Fig. 17).
The experimental data are well described by the model: $I_{inner}/$
$I_{outer} = [I_t \times (t-20) + I_0 \times 20]/(I_0 \times t)$, where $I_0$ is the PL intensity per unit
thickness from the bottom layer and the outer zone of the top layer,
and $I_t$ is the enhanced intensity from the inner zone of the top layer due
to the structural transition (Fig. 4c and Supplementary Note V). The
excellent agreement between the model and experimental data
strongly validates our proposed mechanism, confirming that the
optical heterostructure originates from a transition confined to the top
layer beyond a ~20 nm thickness.

These results collectively confirm that the structural transition is
directly responsible for the optical heterostructure. To further solidify
this causal link, we performed a statistical analysis of the transition
zone and the PL-enhanced inner zone across 120 nanosheets (Sup-
plementary Figs. 18 and 19). The transition zone area (defined by the
initial emergence of both the twin structure and optical hetero-
structure during growth) and the inner zone area (the final bright
central region) were both found to be approximately 5 μm² (Fig. 4d).

Furthermore, during nanosheet growth, the outer zone expands while
the inner zone area remains constant (Supplementary Fig. 20). The
statistical equivalence and constant nature of the inner zone area
provide strong support for the conclusion that the structural transition
defines the region of enhanced emission, thereby establishing it as the
origin of the observed optical heterostructure.

We then sought to elucidate how the structural transition gives
rise to the optical heterostructure phenomenon. It is known that
structural transitions or local disorder can induce exciton localization,
which suppresses exciton diffusion to nonradiative trap sites and
thereby enhances radiative recombination efficiency[36–38]. This
mechanism is consistent with previous reports showing that crystal
packing and order-disorder polytypes in PDBCz modulate lumines-
cence properties[24,39], and nanoscale disorder in organic molecular
solids can induce exciton localization[36,40,41]. To directly probe the
recombination dynamics in our system, we performed time-resolved
photoluminescence (TRPL) measurements. The results show that the
inner zone exhibits a markedly shorter PL lifetime together with a
significantly stronger emission intensity compared to the outer zone
(Fig. 4e). This coexistence of a shorter lifetime and enhanced emission
suggests the introduction of an additional, fast radiative pathway.

Further analysis of the PL decay kinetics reinforces this inter-
pretation. The decay curve in the outer zone is well fitted by a bi-
exponential function with lifetimes $\tau_1 = 0.463$ ns and $\tau_2 = 2.417$ ns. In
contrast, the inner zone requires a tri-exponential function, with life-
times $\tau_1 = 0.467$ ns, $\tau_2 = 2.412$, and an additional, ultrafast component
$\tau_3 = 0.151$ ns. The two longer components (~0.46 ns and ~2.4 ns),

present in both zones, are attributed to the intrinsic recombination channels of PDBCz. The exclusive presence of the ultrafast $\tau_3$ component in the inner zone signifies an extra radiative recombination channel opened by the structural transition. This additional fast pathway accelerates the overall exciton recombination, resulting in the observed shorter lifetime and higher PL efficiency in the inner zone, and conclusively linking the structural transition to the formation of the optical heterostructure.

In conclusion, we report the observation of an intrinsic optical heterostructure and its associated twin structure in a single-component organic nanosheet. This photonic system, characterized by strongly enhanced fluorescence in the inner zone, originates from a spatially localized solid-state transition confined to the upper region—specifically, the top layer beyond ~20 nm from the substrate. This localized restructuring creates a distinct transition region that significantly enhances radiative recombination efficiency, thereby directly generating the observed emission heterogeneity. The entire process is governed by the competitive interplay between molecule-substrate and intermolecular interactions, which evolves as the nanosheet grows.

This work not only establishes a platform for realizing optical heterogeneity in organic materials but also highlights structural dynamics as a fundamental design principle for organic micro-nano photonics. The discovery of a tunable twin structure coupled to optical functionality opens a compelling research direction, calling for further exploration through advanced in-situ characterization and multi-scale theoretical modeling. A deeper understanding of such structure-property relationships will be essential for harnessing structural transitions as a versatile tool for tailoring light-matter interactions. We anticipate that this emerging avenue will not only expand the frontiers of organic material science but also accelerate the development of innovative integrated photonic and optoelectronic devices.

## Methods

### Sample preparation

9H-carbazole (1.0 g, 6.0 mmol) and KOH (0.54 g, 9.6 mmol) were added to a glass flask and dissolved in dimethylformamide (40 mL) at 40 °C, then stirred for 2 hours. 1,4-Dibromo-2,5-difluorobenzene (0.65 g, 2.4 mmol) was introduced into the mixture and heated at 110 °C for 4 hours, followed by cooling to room temperature and filtration. The raw product was purified using silica column chromatography with dichloromethane and petroleum ether as eluents. White powders were obtained with a yield of 60%. A schematic diagram of the synthesis process is shown in Supplementary Fig. 21.

PDBCz nanosheets were synthesized by the physical vapor deposition (PVD) method. In a typical process, PDBCz powder was loaded into an alumina boat and placed at the center of a tube furnace, while a substrate pretreated by O₂ plasma was positioned downstream at a fixed distance from the source. After evacuation to a base pressure of ~0.1 Pa, high-purity Ar gas (99.995%) was introduced as the carrier at a constant flow. The furnace temperature was ramped to 310 °C at a rate of 10 °C min⁻¹ and maintained for 1 hour, followed by natural cooling to room temperature. This setup enabled the reproducible growth of uniform PDBCz nanosheets. Detailed synthetic procedure for PDBCz nanosheets, together with schematic illustrations of the experimental setup, are provided in Supplementary Note VI and Supplementary Fig. 22.

In addition, nanosheets were also obtained by a solution-based self-assembly method, in which controlled solvent evaporation induced molecular ordering on the substrate surface. First, 5.66 mg of PDBCz powder was dispersed in 10 mL of trichloromethane (TCM) and sonicated for 5 minutes to form a homogeneous solution. Subsequently, 1 mL of N,N-dimethylformamide (DMF) and 1 mL of dichloromethane (DCM) were added, and the resulting transparent solution was allowed to stand at room temperature for 3 days. A 20 μL aliquot of the mixed solution was then drop-cast onto a pre-cleaned substrate and placed in a sealed Petri dish saturated with ethanol vapor. Controlled evaporation of the mixed solvent promoted the self-assembly of PDBCz molecules, resulting in the formation of nanosheets on the substrate surface within 2 hours. A schematic illustration of the experimental setup is provided in Supplementary Fig. 23.

### Characterization

Optical and fluorescence images were recorded using a Nikon DS-Ri2 microscope under UV excitation from a mercury lamp (330-380 nm). Photoluminescence (PL) spectra of the nanosheets were measured with a micro-Raman system (Zolix Finder Smart FST2-MPL501-405C1/ WITEC Alpha 300 R) equipped with a 405 nm continuous-wave laser. PDBCz powders and single crystals were characterized by X-ray diffraction (Nexsa G2). AFM, KPFM, and EFM measurements were performed on an Oxford MFP-3D system, and TEM images were obtained with a JEOL ARM200F operated at 200 kV. Detailed PL measurement conditions, including optical setup, excitation parameters, and data acquisition procedures, are provided in Supplementary Note VII and Supplementary Fig. 24.

### First-principles calculations

Structure optimization calculations based on Density Functional Theory (DFT) were carried out using the Vienna Ab initio Simulation Package (VASP)[42–44]. The exchange-correlation potentials were described through the Perdew-Burke-Ernzerhof (PBE) functional within the generalized gradient approximation (GGA) formalism[45]. A plane wave basis set with an energy cutoff of 450 eV was utilized to expand the electronic wave functions, and the criterion for the total energy tolerance was set below $10^{-4}$ eV. Detailed calculations of adsorption energy and growth rate are provided in Supplementary Notes II and III.

## Data availability

The raw data generated in this study have been deposited in the Figshare database under accession code https://doi.org/10.6084/m9.figshare.30581027. The raw data are freely accessible without restrictions, and data can be obtained from Lin Wang upon request.

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

## Acknowledgements

This work is supported by the National Natural Science Foundation of China (Grant nos. 92477123, 52373290 and 62288102 to L. W., 62305158 to Y. L., 12274067 and 92464101 to Chao Zhu, U25A20484 to J. Z., 92477140 to Chongqin Zhu, 22475098 to Z.A.), the Natural Science Foundation of Jiangsu Province (Grant no. BK20230311 to Y. L.), the Open Fund of State Key Laboratory of Infrared Physics (Grant no. SITP-NLIST-ZD-2024-02 to L. W.), the Open Research Fund of Suzhou Laboratory (Grant no. SZLAB-1608-2024-TS019 to Chao Zhu), the Guangdong Basic and Applied Basic Research Foundation (Grant no. 2023A1515011852) and the Shenzhen Natural Science Foundation (Grant No. JCYJ20250604174400001) to X. Y.

## Author contributions

K. Liao, J.Z., and X.Y. contributed equally to this work. L.W., Chao Zhu, and Z.A. jointly supervised this work. J.Z. and L.W. co-wrote the manuscript. L.W. and K. Liao conceived the original idea. Under the supervision of L.W., K. Liao carried out the majority of the experimental work, including nanosheet synthesis, optical and fluorescence microscopy, PL spectroscopy and mapping, thickness-dependent studies, and thermal annealing tests, with assistance from W.X., Z.X., Y.M., and Z.Z. Chao Zhu conducted most TEM and SAED characterizations. Under the supervision of Chao Zhu, D.Z., C.L., and M.S. performed complementary structural characterizations. Z.A. designed the molecular structure of PDBCz. Under the supervision of Z.A., K. Liu coordinated crystal synthesis and the solution-based synthesis of nanosheets. Under the supervision of J.Z., Z.M. performed AFM, KPFM, and EFM measurements. X.Y. performed first-principles calculations. Y.L., X.W., and Chongqin Zhu provided in-depth guidance on both experimental and theoretical aspects of this work.

## Competing interests

The authors declare no competing interests.
