## [Transparent Peer Review file · Nature Communications]

Optical heterostructure in a two-dimensional organic crystal

Corresponding Author: Professor Lin Wang

Version 0:

Reviewer comments:

Reviewer #1

(Remarks to the Author)

Kan Liao et al reported their work on a novel 2D organic crystal that shows built-in optical heterostructure (OH) formed by out of plan twinning formed in the core of the crystal from protected substrate interactions. This is demonstrated in the enhanced photoluminescence (PL) in the inner zone of the crystal. The uniqueness of the work is the creation of OH in a single component system which subsequently eliminates the complication from common practice with lateral arrangement of multiple moieties. This discovery potentially could advance the field of organic optoelectronic devices.

Pros:

1. A novel synthesis that created a single-component organic OH, which overcome the challenge that low dimensional organic crystals are facing: controllability of structural and optical properties due to the flexibility both in-plane and out-of-plane from the weak van der Waals interaction among molecules, as well as the stacking order among layers.
2. Very beautiful images with high resolution, and impressive illustrations.

Cons:

1. Missing a comprehensive review of relevant work. The authors claimed 'Therefore, the realization of OH within organic materials has, to date, been confined to multi-component systems^{3,14,15,16}'. However, this is overstated. For instance, in 2022 single component organic OH was created in a cubic microdisk via self-assembly which showed high contrast of PL between the edge and the body, and with high optical gain [ref. Adv. Optical Mater. 2022, 10, 2101931].
2. In general, this MS lacks in-depth discussion about the mechanisms behind the observations. Also, the submission is incomplete and misses content about the material synthesis and characterization methods. The SI only contains figures with no explanations. It would greatly enhance the scientific merits of the MS if the SI were more self-sufficient.
3. I have multiple specific questions/concerns about the current MS:

(i) There is no description of PL measurements which are critical for the claim.

Question about the illumination method: if the beam size is smaller than the whole crystal (~8 μ m), then it is expected that the center would be brighter than the edge. The authors should show PLQE (not just the PL intensity as in Fig. 1d) normalized by the incident light intensity.

(ii) The authors cited a few references {ref. 31-33} to explain their observation of nonuniformity in PL. However, the material systems in the citations are different from their crystal, the reviewer feels that it is important that they conduct further in-depth analysis to gain more insights about the OH phenomenon. For instance, measuring PLQE of the inner and outer zones, varying the excitation energy and intensity, and mapping the PL using microscopic setup.

(iii) I have a problem with the label in SI fig. 14. From the top drawing, t should be more than 20nm (probably the substrate SiO₂ thickness). Yet, a few of the bottom graphs have t<20nm which is physically impossible, unless t is mislabeled: it should be the thickness of the crystal nanosheet, not the combined thickness. The same labeling appears in Fig. 4b as well. Suggest the authors to double check and make appropriate correction.

(iv) The explanation illustrated in Fig. 3c has unanswered question: if the formation of top layer crystal structure to m₂ orientation is due to the increasing intermolecular interaction, one would expect this to continue as the size grows bigger passing the critical size, yet, the m₂ orientation mysteriously stopped and the rest of the crystal structure is back to m₁ orientation similar to the bottom layer where the strong interaction helps to hold the structure. The authors need to explain more about the 'thermodynamically-driven island growth mode'.

(v) I do not understand the merit of the fitting formula used in Fig. 4b, and how that is a proof that 'This confirms that the enhancement of PL is arising from the structural transition.' , In addition, it looks to me that there is a wide variation of the inner zone area shown in SI Fig. 16, unlike what the authors claimed 'Remarkably, both areas are approximately 5 μ m² (Fig. 4c).'

(vi) Also, I fail to see the connection between simulation in Fig. 2f and the experimental observations. How does the

simulation predict a critical size of the nanosheet when OH occurs? How is the critical size related to the simulated adsorption energy in Fig. 2g? what factors would alter these parameters? Without answers to these questions, this MS remains at the laboratory report level and cannot provide guidance to help advancement of the science in the field of OH. Furthermore, it will be helpful for the community if the authors could elaborate on potential applications of their cool crystal nanosheet.

Reviewer #2

(Remarks to the Author)

The manuscript submitted by Kan Liao et al. reports an intrinsic OH uniquely observed in a 2D nanosheet composed of a single organic component. This discovery is highly intriguing, as it represents the first demonstration of such a phenomenon in purely organic 2D systems. The authors provide a convincing mechanistic explanation, attributing the OH to an island growth process and the presence of a twin structure, both of which are rarely reported in organic nanosheets. These findings not only shed light on the fundamental physics of organic 2D materials but also open new directions for the design of multifunctional optical materials. I therefore strongly recommend publication in Nature Communications after minor revisions. A few clarifications and minor issues should be addressed prior to acceptance.

1. The authors identified a twin structure in the PDBCz organic nanosheets. To facilitate a deeper understanding of its crystallographic features, please provide the molecular packing structures based on the unit cell along the a-, b-, and c-crystallographic axes, respectively.
2. The authors propose that the nanosheet follows an island growth mode. It is recommended to further clarify the relative growth rates and corresponding surface energies of the major crystal facets, in order to gain deeper insight into the growth kinetics.
3. Figure S12 shows that plasma treatment can effectively eliminate the optical heterojunction phenomenon in the upper layer. To further verify the reliability of this method, it is recommended to provide experimental results on thinner samples after plasma treatment.
4. The authors propose that the twin structure transition in the nanosheets originates from a synergistic competition between intermolecular interactions and the interaction between Br atoms and the silicon substrate. To verify the generality of this mechanism, it is recommended to include observations of the OH phenomenon on SiO₂/Si substrates using structurally similar molecules that do not contain Br atoms.
5. On page 6 of the manuscript, the authors provide adsorption energy values for both SiO₂ and BN substrates, along with intermolecular interaction energies. How these values were computationally derived?

Reviewer #3

(Remarks to the Author)

This work reports an optical heterostructure that exhibits different emission intensities. I agree that optical heterostructures composed of single-component organic molecules are rare. However, I cannot recommend acceptance of this work due to the following concerns:

Why do the inner and outer zones exhibit different emission intensities? The authors attribute this to structural transition but fail to provide a clear explanation.

The authors suggest that the formation of the heterostructure is strongly influenced by interactions between the PDBCz nanosheets and substrates, yet no experiments were conducted to investigate these interfacial interactions.

Many key conclusions in the manuscript are speculative and lack direct evidence, which is not scientifically rigorous. The authors should perform more detailed mechanistic studies.

Did the authors examine the emission properties from different sides of the bulk crystals? Is there any variation in emission intensity?

Version 1:

Reviewer comments:

Reviewer #1

(Remarks to the Author)

Additional suggestion for minor revisions:

1. p5 line 112: ‘...sharing an identical emission maximum at ~552 nm’. From the PL mapping, the color of the center is yellow while the outer red, which contradicts the claim. My guess is that the 2D contour map is pseudo color, which indicates intensity rather than the emission color. The authors should add a sentence clarifying this to avoid misunderstanding.
2. regarding the fitting formula for Fig. 4c: suggest adding more clear description to I_t and I₀, to reflect that I_t only occurs for the top layer (with m2 structure).

Reviewer #2

(Remarks to the Author)

I am satisfied with the revision and it could be accepted.

Reviewer #3

(Remarks to the Author)

I appreciate the authors for their efforts on revising the manuscript, and the paper can be accepted as its current form.

Response Letter (Manuscript ID: NCOMMS-25-44606-T)

Revisions and list of changes:

In Main Text:

1. In the author list, we have added Wenheng Xu, Zhongjing Xia, Zilong Mao, and Yan Lv, in recognition of their contributions to the experiments and the mechanistic interpretation in the revised manuscript.
2. The abstract has been revised and optimized, as highlighted in yellow on page 2.
3. The Introduction has been revised to include an updated perspective on the potential future applications of the optical heterostructure phenomenon on page 3, with changes highlighted in yellow.
4. The discussion regarding the formation of optical heterostructures through optical microcavities or multi-component systems has been refined, and additional references have been included (as highlighted in yellow on page 4).
5. We have added subtitles to each section of the Results and Discussion part.
6. The description of the PL mapping has been added to the Results and Discussion (page 5), with changes highlighted in yellow.
7. We have added a discussion on the absorption in the inner and outer zones, as highlighted in yellow on page 5.
8. The explanation for the brighter emission at the edges has been optimized and updated on page 5, with changes highlighted in yellow.
9. The paragraph discussing the exclusion of certain possible mechanisms for the optical heterostructure phenomenon has been optimized and updated on page 5, with changes highlighted in yellow.
10. The discussion regarding the twin structure being oriented along the *b*-axis has been optimized and updated on page 6, with changes highlighted in yellow.
11. The discussion regarding the theoretical calculations for the nanosheets with similar molecular structures but without Br has been updated on page 7 of the main text, with changes highlighted in yellow.
12. In the discussion of the island growth mechanism, we have incorporated theoretical calculations of growth rate and surface energy, as indicated by the yellow-

highlighted text on page 8.

13. In the discussion of our proposed scenario, we have incorporated theoretical calculations to verify that the outer zone of the top layer remains in the m_2 structure while that of the bottom layer retains the m_1 structure, as indicated by the yellow-highlighted text on page 8.
14. The discussion regarding the evidence for the existence of the structural transition confined to the inner zone has been updated and optimized, as shown in the yellow-highlighted text on pages 9-10.
15. In the section discussing the correlation between structural transition and the optical heterostructure phenomenon, we reorganized the logic by first describing that the structural transition region coincides in area with the inner zone of enhanced PL emission, and then explaining why the structural transition leads to the emergence of the optical heterostructure phenomenon, as indicated by the yellow-highlighted text on pages 9-11.
16. The discussion on formula fitting based on our proposed scenario has been revised and updated, as indicated by the yellow-highlighted section on page 10.
17. The discussion on why the structural transition induces the optical heterostructure phenomenon has been updated and optimized, with the addition of a discussion on TRPL data, as indicated by the yellow-highlighted text on pages 10-11.
18. The conclusions section has been revised and optimized, as highlighted in yellow on pages 11-12.
19. The experimental methods have been optimized and updated as indicated by the yellow-highlighted section on pages 12-13.
20. We have added a Data Availability section on page 13.
21. The references have been updated and adjusted accordingly based on the revisions in the main text.
22. We have added an Additional information section on page 17.
23. A PL mapping image has been added as an inset in Fig. 1d, and the corresponding figure caption has been revised accordingly.
24. The transmission and reflection spectra have been added as Fig. 1e, and the corresponding caption has been revised accordingly.

25. The TEM-EDS data have been added as Fig. 1g, and the corresponding caption has been revised accordingly.
26. In Figs. 2f-h, we incorporated simulations of the bonding configurations of Br-free PDCz molecules on SiO₂ and BN substrates, together with recalculated adsorption energies. The corresponding figure captions have been revised accordingly.
27. Theoretical calculations of surface energies and growth rates for different facets of the nanosheets have been added as Fig. 3c. The related figure captions have been revised accordingly.
28. Molecular arrangements before and after structural relaxation has been added as Fig. 3d. The related figure captions have been revised accordingly.
29. The schematic illustration in Fig. 3e has been optimized and modified to more clearly present the structural transition in the inner zone of top layer. The corresponding figure captions have been revised accordingly.
30. Figure 4 has been reorganized. Optical images before and after thermal annealing, as well as KPFM, PFM and AFM images, have been added for as-grown thick samples and thin samples. The plot in Fig. 4d showing the PL intensity ratio between the inner and outer zones as a function of thickness has been modified by adding schematic illustrations of nanosheets with different thicknesses and optimizing the inset. TRPL data have also been included. The corresponding figure captions have been revised accordingly.
31. The figure numbering in the main text has been updated accordingly.

In Supplementary Information Materials:

1. Molecular structure diagrams of m_1 and m_2 oriented along the a , b , and c crystallographic axes have been added as Supplementary Figure 2.
2. A detailed description of the qualitative comparison of PLQE has been added as Supplementary Note I.
3. Supplementary illustrations of the molecular structures of PDCz and mCP molecules (Br-free, structurally similar to PDBCz), along with their optical and fluorescence images, have been added as Supplementary Figure 8.
4. A description of the adsorption energy calculation method has been added as

Supplementary Note II.

5. A description of the surface energy and growth rate calculation method has been added as Supplementary Note III.
6. Molecular arrangements before and after structural relaxation for the alternative configuration have been added as Supplementary Figure 10.
7. A detailed description of the plasma-treated samples, along with a corresponding schematic illustration, has been added as Supplementary Note IV and Supplementary Figure 12.
8. Optical, AFM, and fluorescence images of a nanosheet thinned to 8.7 nm via plasma treatment have been added as Supplementary Figure 14.
9. Optical and fluorescence images of an artificially thinned PDBCz nanosheet (~18 nm) before and after thermal annealing have been added as Supplementary Figure 16.
10. AFM data have been added to Supplementary Figure 17, and the schematic illustration has been optimized and revised.
11. A detailed description of the fitting procedure for the experimental $I_{\text{inner}}/I_{\text{outer}}$ ratio as a function of thickness has been added as Supplementary Note V.
12. The image used to extract the area of the inner enhanced emission region of the nanosheet has been optimized, primarily by adjusting the scale bar so that the central emission area appears consistent in size across the image, as shown in Supplementary Figure 19.
13. A detailed description and schematic illustration of the PDBCz powder synthesis have been added as Supplementary Note VI and Supplementary Figure 21.
14. A detailed description of the PDBCz nanosheet synthesis via the PVD method has been added, and the corresponding schematic illustration has been revised and optimized. These have been included as Supplementary Note VII and Supplementary Figure 22.
15. A detailed description of the PDBCz nanosheet synthesis via the solution method, along with a corresponding schematic illustration, has been added as Supplementary Note VIII and Supplementary Figure 23.
16. A detailed description of the PL measurements, along with a corresponding

schematic illustration, has been added as Supplementary Note IX and Supplementary Figure 24.

17. All figure captions in the Supplementary Information have been revised to provide detailed descriptions.
18. The figure numbering in the Supplementary Information has been updated to align with the revised main text.

Reviewer #1' comments:

Kan Liao *et al.* reported their work on a novel 2D organic crystal that shows built-in optical heterostructure (OH) formed by out of plane twinning formed in the core of the crystal from protected substrate interactions. This is demonstrated in the enhanced photoluminescence (PL) in the inner zone of the crystal. The uniqueness of the work is the creation of OH in a single component system which subsequently eliminates the complication from common practice with lateral arrangement of multiple moieties. This discovery potentially could advance the field of organic optoelectronic devices.

Pros:

1. A novel synthesis that created a single-component organic OH, which overcome the challenge that low dimensional organic crystals are facing: controllability of structural and optical properties due to the flexibility both in-plane and out-of-plane from the weak van der Waals interaction among molecules, as well as the stacking order among layers.

2. Very beautiful images with high resolution, and impressive illustrations.

Replies to Reviewer #1's Comments:

We greatly appreciate Reviewer #1's positive and encouraging assessment of our work. We are delighted that the novelty of constructing a single-component organic optical heterostructure, as well as the quality of our experimental data and visualizations, have been recognized. Guided by the reviewer's insightful comments, we have carefully revised and refined the manuscript to further enhance its clarity, coherence, and overall presentation. Below, we provide detailed, point-by-point replies to each comment.

Comment 1:

Missing a comprehensive review of relevant work. The authors claimed 'Therefore, the realization of OH within organic materials has, to date, been confined to multi-component systems [3,14,15,16]. However, this is overstated. For instance, in 2022 single component organic OH was created in a cubic microdisk via self-assembly which showed high contrast of PL between the edge and the body, and with high optical gain [ref. Adv. Optical Mater. 2022, 10, 2101931].

Reply to Comment 1:

We sincerely appreciate the reviewer's careful reading and for pointing out the relevant report in ref. [Adv. Optical Mater. 2022, 10, 2101931], for which we have made the description more rigorous in our revised manuscript. The optical heterostructure phenomenon described in that work is fundamentally different from the optical heterostructure observed in our study. To facilitate a clearer understanding of the distinct luminescent regions and mechanisms in the optical heterostructure, we have included simplified schematic diagrams in Figure R1.1. In ref. [Adv. Optical Mater. 2022, 10, 2101931], the emission contrast appears at the very edges of cubic microdisks and is primarily attributed to the optical waveguide and whispering-gallery-mode (WGM) effects. By contrast, our nanosheets exhibit an intrinsic optical heterostructure within the central region, where the inner zone shows significantly stronger photoluminescence than the surrounding outer zone. This phenomenon cannot be explained by cavity or waveguide effects but instead originates from a unique out-of-plane twin structural transition driven by the interplay between molecule-substrate and intermolecular interactions.

Figure R1.1. The top panels present schematic illustrations of the luminescent regions and mechanisms in the o-MSB microdisk [Adv. Optical Mater. 2022, 10, 2101931] (a) and the PDBCz nanosheet in our work (b), while the bottom panels show their corresponding fluorescence images.

Notably, in addition to this optical heterostructure phenomenon with significantly

stronger photoluminescence than the surrounding outer zone, our nanosheets also display brighter outer edges due to the optical waveguide effect. Accordingly, ref. [Adv. Optical Mater. 2022, 10, 2101931] has been cited again in the main text within the sentence discussing that “It is also noteworthy that the outer edge can exhibit brighter emission due to optical confinement effects like waveguide modes, which is distinct from the optical heterostructure reported here.”

Comment 2:

In general, this MS lacks in-depth discussion about the mechanisms behind the observations. Also, the submission is incomplete and misses content about the material synthesis and characterization methods. The SI only contains figures with no explanations. It would greatly enhance the scientific merits of the MS if the SI were more self-sufficient.

Reply to Comment 2:

We sincerely thank the reviewer for this valuable and constructive comment. In the revised manuscript, we have added detailed descriptions of the material synthesis and characterization methods in the main text to ensure completeness. A more comprehensive version of these methods, together with additional datasets, has been provided in the Supplementary Information (Supplementary Notes IV, VI-IX; Supplementary Figures 4 and 21-24). Furthermore, all figures in the Supplementary Information are now accompanied by detailed captions to improve clarity, consistency, and scientific rigor.

The observed optical heterostructure phenomenon is originating from a structural transition, as accompanied with the emergence of twin structures. To address the reviewer’s concern and provide a clearer mechanistic framework, we have reorganized the manuscript to present a concise logical progression of the underlying mechanism, with key experimental evidence summarized in Tables R1.1-1.5.

The logic of our work can be outlined as follows:

(1) **Interfacial Interaction Dominance:** At the early growth stage, strong molecule-substrate interactions, particularly those involving the Si-Br bond, dominate molecular packing and lead to the formation of uniform m_1 lattice.

(2) **Structural Transition and Twin Formation:** As the nanosheet laterally expands,

the molecule-substrate interaction becomes less dominant compared to the intermolecular interaction, triggering a structural transition in the top layer from m_1 to m_2 , forming an out-of-plane twin structure ($m_1 + m_2$).

(3) **Optical Consequence:** The structural transition area exhibits a new recombination pathway (~ 0.15 ns), resulting in enhanced photoluminescence efficiency in the inner region.

In summary, the structural transition (Tables R1.1-1.2) \rightarrow twin formation (Table R1.1) and PL enhancement (Table R1.3) \rightarrow optical heterostructure (Table R1.4) sequence provides a coherent explanation for the observed phenomena. We also carried out a series of material characterizations aimed at excluding simple and conventional explanations (Table R1.5). To further substantiate this mechanism, we have incorporated additional PL mapping, KPFM/EFM measurements, transmittance/reflectance spectra, TRPL data, and theoretical calculations in the revised manuscript (highlighted in yellow in the main text). Moreover, we have expanded the Supplementary Information with detailed synthesis procedures, characterization methods, and explanatory notes to make it fully self-contained and scientifically rigorous.

Table R1.1 Structural transition and twin formation induced by molecule-substrate interfacial interactions originating from Si-Br bonding.

Observations /Experiments	Descriptions	Figures
PDBCz on siliceous substrates	Twin structure and OH	Figure 1c and Supplementary Figure 7
PDBCz on non-siliceous substrates	One set diffraction and non-OH	Figure 2e and Supplementary Figure 6
Similar molecular without Br on siliceous substrates	Non-OH	Supplementary Figure 8
PDBCz on siliceous substrates by solution method	One set diffraction and non-OH	Figure 2d and Supplementary Figure 11

Table R1.2 Existence of the structural transition.

Observations /Experiments	Descriptions	Figures
Thermal annealing	Transition zone is more sensitive to heat	Figure 4a
KPFM and EFM	Crystallinity difference between inner and outer zones induced by the structural transition	Figure 4a
Plasma-thinned and as-grown thin nanosheet	Thick and thin (< 20 nm) samples show totally different behaviors in most characterizations, suggesting the structural transition only appears at the top layer of inner zone	Figure 4b and Supplementary Figure 13-15

Table R1.3 Structural-transition-induced PL enhancement in the inner zone.

Observations /Experiments	Descriptions	Figures
I_{inner}/I_{outer} vs t	The trend is in good agreement with our proposed scenario	Figure 4c
Sizes of transition and inner zones	Transition zone is consisted with the PL-enhanced inner zone	Figure 4d
Plasma-thinned and as-grown thin nanosheet	Simultaneous disappearance of transition and PL-enhancement in the inner zone for both types of thin samples	Figure 4b and Supplementary 13-15
TR-PL	The additional fast radiative process in the transition zone leads to enhanced PL	Figure 4e

Table R1.4 Optical heterostructure.

Observations /Experiments	Descriptions	Figures
Fluorescence images	Brighter inner zone than outer zone	Figure 1c
PL spectra	Stronger PL in the inner zone	Figure 1d
PLQE	Higher PLQE in the inner zone	Supplementary Note I
PL Mapping	Stronger PL in the inner zone	Figure 1d

Table R1.5 Exclusion of possible mechanisms for the OH phenomenon.

Observations /Experiments	Descriptions	Figures
Optical images	No visible difference between inner and outer zones	Figure 1c
AFM images	No thickness difference between inner and outer zones	Figure 1e
TEM-EDS	Uniform elemental distribution in the inner and outer zones	Figure 1g
TEM-SAED	Highly consistent crystal structures of the inner and outer zones	Figure 2a
Optical microcavity effect	OH persists despite disrupted nanosheet shape	Supplementary Figure 3

Comment 3:

I have multiple specific questions/concerns about the current MS:

(i) There is no description of PL measurements which are critical for the claim. Question about the illumination method: if the beam size is smaller than the whole crystal (~8 μm), then it is expected that the center would be brighter than the edge. The authors should show PLQE (not just the PL intensity as in Fig. 1d) normalized by the incident light intensity.

Reply to Comment 3:

We sincerely thank the reviewer for this valuable comment. Photoluminescence (PL) spectra and mapping of the nanosheets were performed using a micro-Raman system (Zolix Finder Smart FST2-MPL501-405C1 or WITec Alpha 300R). The substrate was mounted under a confocal optical microscope, where a 405 nm continuous-wave laser was focused onto the sample surface through a dry objective lens (NA = 0.55), yielding a $\sim 1 \mu\text{m}$ spot size. All measurements were conducted at room temperature. The PL was collected by the same objective, filtered through a 405 nm long-pass filter, and directed to a spectrometer equipped with a CCD detector, with a typical integration time of 1 s. The corresponding optical path is illustrated in Figure R1.2 (also see Supplementary Note IX and Supplementary Figure 24). Fluorescence images were recorded on a Nikon DS-Ri2 microscope camera under UV excitation from a mercury lamp (Nikon INTENSILIGHT C-HGFI) with a 330-380 nm bandpass filter.

Figure R1.2 Schematic diagram of PL measurement setup.

We also sincerely thank the reviewer for raising this important point regarding the direct measurement of the PLQE for the inner and outer zones. We fully agree that quantifying the absolute PLQE would provide a more direct measure of the emission efficiency difference.

However, in our experimental context, performing a direct and accurate micro-scale PLQE measurement presents significant challenges. The established and most reliable method for absolute PLQE determination is the integrating sphere approach^{1, 2}. This technique, however, is inherently a macro-scale measurement that collects and averages the total photon flux from the entire sample area illuminated by the excitation beam. It lacks the spatial resolution to isolate and compare signals from two distinct sub-micron regions within a single nanosheet, particularly given that the inner zone's dimensions are confined to under 2 μm .

Unfortunately, after an extensive search, we have not been able to identify a commercially available or custom-built system that can perform direct and absolute PLQE measurements with the required sub-2 μm spatial resolution on our samples.

Given this technical limitation, we employed a series of indirect but compelling qualitative and semi-quantitative experiments to robustly compare the relative emission efficiency between the zones:

(1) Spatially resolved PL spectroscopy and mapping:

Using a confocal micro-PL system, we separately acquired PL spectra and performed PL mapping for the inner and outer zones (Figure 1d). The PL intensity in the inner zone is consistently and substantially stronger than that in the outer zone, confirming pronounced emission heterogeneity at the microscale.

(2) Comparison of optical absorption:

To further determine whether the difference in PL intensity originates from variations in absorption, we measured the reflectance and transmittance spectra of both regions. As shown in Figure 1e of the main text, the two spectra are nearly identical, indicating negligible differences in optical absorption and thus comparable photon excitation conditions between the inner and outer zones³.

Since the absorption is comparable while the PL emission is markedly stronger in the inner zone, we can qualitatively conclude that its PLQE is significantly higher than that of the outer zone.

Comment 4:

(ii) The authors cited a few references (ref. 31-33) to explain their observation of

nonuniformity in PL. However, the material systems in the citations are different from their crystal, the reviewer feels that it is important that they conduct further in-depth analysis to gain more insights about the OH phenomenon. For instance, measuring PLQE of the inner and outer zones, varying the excitation energy and intensity, and mapping the PL using microscopic setup.

Reply to Comment 4:

We thank the reviewer for this valuable comment. We agree that it is important to provide literature evidence more closely related to our material system. In the revised manuscript, we have added relevant studies on PDBCz itself and structurally analogous carbazole derivatives. For example, Shi *et al.* reported that PDBCz exhibits highly efficient ultralong phosphorescence whose efficiency is strongly affected by crystal packing and local structural motifs⁴. Moreover, crystallographic studies revealed that PDBCz crystals adopt order-disorder polytypes, which highlights the intrinsic presence of structural heterogeneity in this system⁵. These works directly support our conclusion that local structural transitions and disorder can influence exciton dynamics in PDBCz.

Furthermore, we have cited additional reports on exciton transport in organic molecular solids, which demonstrate that structural disorder leads to exciton localization, and thereby enhances radiative recombination⁶⁻⁸. These studies provide a broader mechanistic foundation for understanding the suppression of exciton annihilation in disordered molecular crystals.

To address the comment regarding PL quantification, we have supplemented our study with micro-PL mapping data. This provides direct spatial visualization of the emission heterogeneity, which we have now included as an inset in Figure 1d of the revised manuscript. Furthermore, as detailed in our response above, we have thoroughly explained the challenges associated with direct micro-scale PLQE measurements and the compelling indirect evidence supporting the higher radiative efficiency in the inner zone.

Comment 5:

(iii) I have a problem with the label in SI Fig. 14. From the top drawing, t should be more than 20 nm (probably the substrate SiO₂ thickness). Yet, a few of the bottom graphs have $t < 20$ nm which is physically impossible, unless t is mislabeled: it should

be the thickness of the crystal nanosheet, not the combined thickness. The same labeling appears in Fig. 4b as well. Suggest the authors to double check and make appropriate correction.

Reply to Comment 5:

We sincerely thank the reviewer for this careful observation. As illustrated in the schematic diagram of Figure R1.3, a nanosheet exhibiting the optical heterostructure phenomenon comprises a top layer with m_2 orientation (blue) and a bottom layer with m_1 orientation (pink). The total thickness of the nanosheet is denoted as t . For nanosheets that display the optical heterostructure phenomenon, t exceeds 20 nm, with the bottom layer maintained at ~ 20 nm and the top layer corresponding to $(t - 20)$ nm. In contrast, nanosheets with a total thickness below 20 nm lack the top structural component, and their entire thickness consists solely of the m_1 orientation. Consequently, as shown in Supplementary Figures 13-16, nanosheets thinner than 20 nm do not exhibit the optical heterostructure phenomenon.

Figure R1.3 The schematic diagram of a PDBCz nanosheet with thickness t , exhibiting the optical heterostructure phenomenon, consisting a top layer with m_2 orientation (blue, $t - 20$ nm) and a bottom layer with m_1 orientation (pink, ~ 20 nm).

To avoid potential misinterpretation, the schematic diagrams in Figure 4c in the main text and Supplementary Figure 17 have been updated to explicitly indicate the total thickness t on the right side. In addition, the corresponding AFM images for each nanosheet shown in Supplementary Figure 17 have been included in the revised SI.

Comment 6:

(iv) The explanation illustrated in Fig. 3c has unanswered question: if the formation of top layer crystal structure to m_2 orientation is due to the increasing intermolecular interaction, one would expect this to continue as the size grows bigger passing the critical size, yet, the m_2 orientation mysteriously stopped and the rest of the crystal structure is back to m_1 orientation similar to the bottom layer where the strong interaction helps to hold the structure. The authors need to explain more about the ‘thermodynamically-driven island growth mode’.

Reply to Comment 6:

We appreciate the reviewer’s insightful question. Through in-situ and real-time investigations of the optical morphology of PDBCz nanosheets at different growth stages, we found that the process follows an island growth mode (top of Figure 3a). Under this mode, the nanosheets undergo predominant lateral expansion with only minimal variation in thickness throughout growth.

To obtain deeper insights into the growth kinetics, we performed theoretical calculations to elucidate the relative growth rates and surface energies of the major crystal facets. The theoretical calculations were performed using *Materials Studio 2017*, where the growth and equilibrium morphologies of PDBCz were evaluated with the Forcite energy method. The simulations employed the universal force field with medium accuracy. Experimentally, monoclinic PDBCz was identified to crystallize in the space group $P2_1/c$, with lattice parameters $a = 8.710 \text{ \AA}$, $b = 17.149 \text{ \AA}$, $c = 8.657 \text{ \AA}$, $\alpha = \gamma = 90^\circ$, and $\beta = 116.339^\circ$, in good agreement with those used in our calculations⁹.

As shown in Figure R1.4a, PDBCz nanosheet can adopt two growth orientations on substrates: perpendicular (standing-up) or parallel (lying-down). To elucidate the growth mechanism and packing mode, both growth and equilibrium morphology methods were employed. The growth morphology was analyzed using the attachment energy method, which relates the growth rate of a given surface to the potential energy per unit cell released upon the addition of a new layer in vacuum. This method provides insights into crystal habits under non-equilibrium growth conditions. The attachment energy (E_{att}) is defined as the energy released when a growth slice attaches to the crystal surface¹⁰. It can be calculated as¹¹:

$$E_{\text{att}} = E_{\text{latt}} - E_{\text{slice}}$$

where E_{latt} is lattice energy of the crystal and E_{slice} is the energy of a growth slice with thickness equal to the interplanar distance. The growth rate of a crystal facet is assumed to be proportional to its attachment energy: facets with the lowest E_{att} grow most slowly and thus dominate the morphology. In contrast, the equilibrium morphology method considers that, under equilibrium conditions, crystal habits minimize the total surface free energy¹². The surface free energy not only governs the crystal growth process but also determines the molecular orientation in the lattice. Indeed, organic molecular packing generally follows the principle of minimizing total surface free energy¹³.

As shown in Figure R1.4b, our calculations of the (100), (011), and (020) facets reveal that the (020) surface exhibits the lowest surface free energy, corresponding to the slowest growth rate and hence the largest exposed facet. This result is in excellent agreement with experimental observations in our work. The small difference between the surface free energies of the (100) and (011) planes indicates comparable growth rates, reflecting the slight anisotropy in the lattice parameters along these directions. Overall, the horizontal growth model, with the crystal oriented parallel to the substrate, minimizes the system's surface energy and thus represents the most favorable growth configuration.

Figure R1.4 (a) The crystal growth model, where the arrows denote the preferred growth orientations. (b) The calculated surface free energies and corresponding growth rates of the (100), (011), and (020) facets.

Experimentally, through real-time optical images captured at different time intervals, it is evident that the lateral dimensions of the nanosheets increase significantly over time (Figure 3a in the main text). Although atomic force microscopy (AFM) measurements could not be performed during the dynamic growth process, the

consistent coloration observed in the optical images suggests that the thickness of the nanosheets remains largely unchanged. This observation further corroborates the theoretical calculations.

To elucidate why the outer zone of the top layer remains in the m_2 orientation during lateral growth after the structural transition, rather than reverting to m_1 , we carried out theoretical structural relaxation simulations (Figure R1.5). When m_1 (or m_2) molecules are placed positioned adjacent m_2 (or m_1) molecules within the lateral a - c plane, relaxation drives the system toward a uniform molecular orientation and ordered packing, confirming that heterogeneous m_1 - m_2 arrangements within the lateral a - c plane are energetically unfavorable and spontaneously eliminated.

Figure R1.5 Molecular arrangements before and after structural relaxation. When m_1 (or m_2) molecules are placed positioned adjacent m_2 (or m_1) molecules within the lateral a - c plane, relaxation drives the system toward a uniform molecular orientation and ordered packing, confirming that heterogeneous m_1 - m_2 arrangements within the lateral a - c plane are energetically unfavorable and spontaneously eliminated.

Comment 7:

(v) I do not understand the merit of the fitting formula used in Fig. 4b, and how that is a proof that ‘This confirms that the enhancement of PL is arising from the structural

transition.’ In addition, it looks to me that there is a wide variation of the inner zone area shown in SI Fig. 16, unlike what the authors claimed ‘Remarkably, both areas are approximately $5 \mu\text{m}^2$ (Fig. 4c).’

Reply to Comment 7:

We appreciate the reviewer’s insightful comment on the fitting formula used in Figure 4b (Figure 4c in the revised manuscript). In our experiments, the PL spectra of the inner and outer zones were recorded under 405 nm excitation, which can penetrate and excite the bottom layer¹⁴⁻¹⁶. The excitation spot size was $\sim 1 \mu\text{m}^2$, smaller than the inner zone area ($\sim 5 \mu\text{m}^2$). We define the PL intensity per unit thickness of the bottom layer and the outer zone of the top layer as I_0 , and that of the inner zone of the top layer as I_t , which differs from I_0 due to structural transition.

Accordingly, the PL intensity of the inner zone (I_{inner}) can be expressed as:

$$I_{\text{inner}} = [I_t \times (t - 20) + I_0 \times 20] \times (\text{excitation spot size})$$

while that of the outer zone (I_{outer}) is:

$$I_{\text{outer}} = (I_0 \times t) \times (\text{excitation spot size}).$$

Thus, the ratio can be written as :

$$I_{\text{inner}}/I_{\text{outer}} = [I_t \times (t - 20) + I_0 \times 20]/(I_0 \times t)$$

Using this relation, the experimental data of $I_{\text{inner}}/I_{\text{outer}}$ as a function of nanosheet thickness (t) were fitted, as shown in Figure 4c. The excellent agreement between experiment and fitting confirms the validity of the model.

We apologize for the misunderstanding caused by Supplementary Figure 19 in revised version, where the non-uniform scale bars created the impression of a large variation in the inner zone. In the revised version, we have standardized the scale bars to ensure consistency.

Comment 8:

(vi) Also, I fail to see the connection between simulation in Fig. 2f and the experimental observations. How does the simulation predict a critical size of the nanosheet when OH occurs? How is the critical size related to the simulated adsorption energy in Fig. 2g? what factors would alter these parameters? Without answers to these questions, this MS remains at the laboratory report level and cannot provide guidance

to help advancement of the science in the field of OH. Furthermore, it will be helpful for the community if the authors could elaborate on potential applications of their cool crystal nanosheet.

Comment 8:

We appreciate the reviewer's insightful question. Figure 2f presents the theoretical simulations of a PDBCz molecule on SiO₂ and BN substrates to evaluate their adsorption energies. The adsorption energy of PDBCz-SiO₂ system (-0.764 eV) is lower than both its intermolecular interaction energy (-0.723 eV) and that of the PDBCz-BN system (close to 0 eV). This reveals that the PDBCz-SiO₂ interface possesses markedly stronger binding strength, primarily induced by Si-Br bonding. The critical size is defined as the characteristic nanosheet dimension at which a single- to twin-crystalline structural transition occurs. This transition arises when the influence of interfacial interactions becomes less dominant than intermolecular interactions as the nanosheet grows larger. We regret that the theoretical calculations cannot be performed at this "critical size" scale of several micrometers, as the computational cost would be prohibitively high.

However, we analyzed a large number of PDBCz nanosheets to identify the onset of the optical heterostructure phenomenon and twin structure during the growth process to obtain the critical size (Supplementary Figure 18). The critical size is approximately the same across samples, indicating that it is not altered by other factors.

Also, we find that, while varying the initial orientation or position of PDBCz molecules relative to either the substrate or another PDBCz molecule may lead to quantitative differences in the calculated adsorption or interaction energies, the qualitative conclusions remain unchanged. The fundamental physical behaviors we observed are robust against such variations in initial configurations.

The demonstrated nanosheets with intrinsic optical heterostructure open up several potential application opportunities. First, the distinct PL contrast between the inner and outer zones can be exploited for high-resolution optical coding, data encryption, and anticounterfeiting, where spatially confined optical responses are highly desirable. Second, the enhanced emission efficiency in the inner zone offers promising prospects for light-emitting devices and integrated photonic components, in particular for microscale lasers and optical waveguides, where locally regulated luminescence is

critical. Third, the well-defined structural transition underlying the optical heterostructure provides a new platform for constructing logic elements and optoelectronic circuits that utilize spatially separated emissive regions as functional units. Finally, the simple single-component nature and vapor-phase growth compatibility of these nanosheets make them attractive for scalable integration into flexible and multifunctional organic optoelectronic devices. We therefore believe that the unique structural-optical correlation uncovered here will inspire new design strategies for advanced organic materials and their device applications.

Reviewer #2' comments:

The manuscript submitted by Kan Liao *et al.* reports an intrinsic OH uniquely observed in a 2D nanosheet composed of a single organic component. This discovery is highly intriguing, as it represents the first demonstration of such a phenomenon in purely organic 2D systems. The authors provide a convincing mechanistic explanation, attributing the OH to an island growth process and the presence of a twin structure, both of which are rarely reported in organic nanosheets. These findings not only shed light on the fundamental physics of organic 2D materials but also open new directions for the design of multifunctional optical materials. I therefore strongly recommend publication in *Nature Communications* after minor revisions. A few clarifications and minor issues should be addressed prior to acceptance.

Replies to Reviewer #2's Comments:

We sincerely appreciate Reviewer #2's positive evaluation, recognizing that our findings not only provide new insights into the fundamental physics of organic 2D materials but also open promising avenues for the design of multifunctional optical materials. In response to the reviewer's constructive comments, we have carefully revised and improved the manuscript. Below, we provide detailed, point-by-point replies to each comment.

Comment 1:

The authors identified a twin structure in the PDBCz organic nanosheets. To facilitate a deeper understanding of its crystallographic features, please provide the molecular packing structures based on the unit cell along the *a*-, *b*-, and *c*-crystallographic axes, respectively.

Reply to Comment 1:

We appreciate the reviewer's insightful suggestion. Figure R2.1 presents the molecular packing structures of the unit cell viewed along the *a*-, *b*-, and *c*-crystallographic axes. The white, gray, blue, and brown spheres correspond to hydrogen, carbon, nitrogen, and bromine atoms, respectively. The three colored arrows denote the crystallographic axes *a*, *b*, and *c*, while the black solid lines mark the boundary of a single unit cell. The PDBCz crystal belongs to the $P2_1/c$ space group, with lattice parameters of $a = 8.710 \text{ \AA}$, $b = 17.149 \text{ \AA}$, $c = 8.657 \text{ \AA}$, $\alpha = \gamma = 90^\circ$, and $\beta = 116.339^\circ$.

Figure R2.1 The molecular packing structures of the unit cell viewed along the a -, b -, and c -crystallographic axes for m_1 (a-c) and m_2 (d-f) structure. The white, gray, blue, and brown spheres correspond to hydrogen, carbon, nitrogen, and bromine atoms, respectively. The three colored arrows denote the crystallographic axes a , b , and c , while the black solid lines mark the boundary of a single unit cell. The PDBCz crystal belongs to the $P2_1/c$ space group, with lattice parameters of $a = 8.710 \text{ \AA}$, $b = 17.149 \text{ \AA}$, $c = 8.657 \text{ \AA}$, $\alpha = \gamma = 90^\circ$, and $\beta = 116.339^\circ$.

Comment 2:

The authors propose that the nanosheet follows an island growth mode. It is recommended to further clarify the relative growth rates and corresponding surface energies of the major crystal facets, in order to gain deeper insight into the growth kinetics.

Reply to Comment 2:

We appreciate the reviewer's valuable suggestion. To obtain deeper insights into the growth kinetics, we performed theoretical calculations to elucidate the relative growth rates and surface energies of the major crystal facets.

The theoretical calculations were performed using *Materials Studio 2017*, where the

growth and equilibrium morphologies of PDBCz were evaluated with the Forcite energy method. The simulations employed the universal force field with medium accuracy. Experimentally, monoclinic PDBCz was identified to crystallize in the space group $P2_1/c$, with lattice parameters $a = 8.710 \text{ \AA}$, $b = 17.149 \text{ \AA}$, $c = 8.657 \text{ \AA}$, $\alpha = \gamma = 90^\circ$, and $\beta = 116.339^\circ$, in good agreement with those used in our calculations⁹.

As shown in Figure R2.2 **a**, PDBCz nanosheet can adopt two growth orientations on substrates: perpendicular (standing-up) or parallel (lying-down). To elucidate the growth mechanism and packing mode, both growth and equilibrium morphology methods were employed. The growth morphology was analyzed using the attachment energy method, which relates the growth rate of a given surface to the potential energy per unit cell released upon the addition of a new layer in vacuum. This method provides insights into crystal habits under non-equilibrium growth conditions. The attachment energy (E_{att}) is defined as the energy released when a growth slice attaches to the crystal surface¹⁰. It can be calculated as¹¹:

$$E_{\text{att}} = E_{\text{latt}} - E_{\text{slice}}$$

where E_{latt} is lattice energy of the crystal and E_{slice} is the energy of a growth slice with thickness equal to the interplanar distance. The growth rate of a crystal facet is assumed to be proportional to its attachment energy: facets with the lowest E_{att} grow most slowly and thus dominate the morphology. In contrast, the equilibrium morphology method considers that, under equilibrium conditions, crystal habits minimize the total surface free energy¹². The surface free energy not only governs the crystal growth process but also determines the molecular orientation in the lattice. Indeed, organic molecular packing generally follows the principle of minimizing total surface free energy¹³.

As shown in Figure R2.2 **b**, our calculations of the (100), (011), and (020) facets reveal that the (020) surface exhibits the lowest surface free energy, corresponding to the slowest growth rate and hence the largest exposed facet. This result is in excellent agreement with experimental observations in our work. The small difference between the surface free energies of the (100) and (011) planes indicates comparable growth rates, reflecting the slight anisotropy in the lattice parameters along these directions. Overall, the horizontal growth model, with the crystal oriented parallel to the substrate, minimizes the system's surface energy and thus represents the most favorable growth configuration.

Figure R2.2 (a) The crystal growth model, where the arrows denote the preferred growth orientations. (b) The calculated surface free energies and corresponding growth rates of the (100), (011), and (020) facets. The (020) surface exhibits the lowest surface free energy, corresponding to the slowest growth rate and hence the largest exposed facet. Overall, the island growth model, with the crystal oriented parallel to the substrate, minimizes the system's surface energy and thus represents the most favorable growth configuration.

Comment 3:

Figure S12 shows that plasma treatment can effectively eliminate the optical heterojunction phenomenon in the upper layer. To further verify the reliability of this method, it is recommended to provide experimental results on thinner samples after plasma treatment.

Reply to Comment 3:

We appreciate the reviewer's valuable suggestion. As shown in the top row of Figure R2.3, a PDBCz nanosheet with thickness of 204.5 nm exhibits optical heterostructure phenomenon. After plasma etching, the nanosheet was thinned to 8.7 nm, which is even thinner than the 17.8 nm sample discussed in the main text. As shown in the bottom row of Figure R2.3, the optical heterostructure phenomenon disappears in the etched nanosheet, confirming that plasma treatment can effectively eliminate the optical heterostructure phenomenon in the upper layer.

Figure R2.3 The optical, AFM and fluorescence images of a PDBCz nanosheet before (the top row) and after (the bottom row) plasma etching treatment.

Comment 4:

The authors propose that the twin structure transition in the nanosheets originates from a synergistic competition between intermolecular interactions and the interaction between Br atoms and the silicon substrate. To verify the generality of this mechanism, it is recommended to include observations of the OH phenomenon on SiO₂/Si substrates using structurally similar molecules that do not contain Br atoms.

Reply to Comment 4:

We appreciate the reviewer's valuable suggestion. We have synthesized 1,4-di(-H-carbazol-9-yl)benzene (PDCz)⁴ and 1,3-bis(N-carbazolyl)benzene (mCP)¹⁷ nanosheets as shown in Figure R2.4, whose molecular structures are similar to that of PDBCz but without bromine atoms. Specifically, PDCz adopts a para-carbazole configuration, whereas mCP adopts an ortho-carbazole configuration. As shown in their fluorescence images, neither material exhibits the optical heterostructure phenomenon. Theoretical calculations also indicate that the PDCz-SiO₂ system (−0.443 eV) exhibits a weaker molecule-substrate interaction compared with its intermolecular interaction energy (−1.071 eV), suggesting that the interfacial interaction is weaker than the

intermolecular coupling (Figure 2h in the main text).

Figure R2.4 The molecular structure, optical and fluorescence images of PDCz (the top row) and mCP (the bottom row) nanosheet on SiO₂/Si substrates.

Comment 5:

On page 6 of the manuscript, the authors provide adsorption energy values for both SiO₂ and BN substrates, along with intermolecular interaction energies. How these values were computationally derived?

Reply to Comment 5:

We thank the reviewer for the question. Regarding the first-principles calculations, we employed the VASP software, with detailed computational methodologies already described in the Method of main text. For the adsorption energy calculations, we strictly followed the formula:

$$E_{\text{ads}} = E_{\text{total}} - E_{\text{PDBCz}} - E_{\text{sub}}$$

where E_{ads} gives the adsorption energy, E_{total} represents the energy of a single PDBCz molecule adsorbed on the substrate, E_{PDBCz} denotes the energy of an isolated PDBCz molecule, and E_{sub} is the energy of the bare substrate. Both pre-adsorption and post-adsorption structures were fully relaxed, with the detailed numerical results presented in Table R2.1.

Regarding the intermolecular interactions between PDBCz molecules, we calculated: 1) The energy of two isolated PDBCz molecules ($-359.936 \times 2 = -719.872$ eV), 2) The

energy of two PDBCz molecules interacting with each other (-720.595 eV). The intermolecular interaction energy was then obtained by taking the difference between these values (-0.723 eV).

While regarding the intermolecular interactions between PDCz molecules, we calculated: 1) The energy of two isolated PDCz molecules ($-363.863 \times 2 = -727.726$ eV), 2) The energy of two PDCz molecules interacting with each other (-728.797 eV). The intermolecular interaction energy was then obtained by taking the difference between these values (-1.071 eV).

Table R2.1 The calculated adsorption energy for both SiO₂ and BN substrates.

	PDBCz@SiO ₂	PDBCz@BN	PDCz@SiO ₂	PDCz@BN
$E_{\text{PDBCz}}/E_{\text{PDCz}}$ (eV)	-356.231	-356.143	-360.711	-360.711
E_{sub} (eV)	-115.293	-1425.022	-115.293	-1425.022
E_{total} (eV)	-472.288	-1781.166	-476.447	-1785.733
E_{ads} (eV)	-0.764	-0.001	-0.443	0

Reviewer #3' comments:

This work reports an optical heterostructure that exhibits different emission intensities. I agree that optical heterostructures composed of single-component organic molecules are rare. However, I cannot recommend acceptance of this work due to the following concerns.

Replies to Reviewer #3' comments:

We sincerely thank Reviewer #3 for acknowledging the rarity of realizing optical heterostructures in single-component organic systems. We fully understand the reviewer's concerns regarding the mechanistic explanation and experimental evidence. In the revised manuscript, we have substantially strengthened the work to address these points.

Comment 1:

Why do the inner and outer zones exhibit different emission intensities? The authors attribute this to structural transition but fail to provide a clear explanation.

Reply to Comment 1:

We sincerely thank the reviewer for raising this important question. The distinct emission intensities between the inner and outer zones arise from a structural transition confined to the inner zone in the top layer of the nanosheet.

1). Evidence of structural-transition-induced optical heterostructure.

1) From direct experimental observations, the region undergoing the structural transition precisely coincides with the area exhibiting enhanced photoluminescence at the nanosheet center. Quantitatively, the area of the transition zone—defined as the nanosheet region where both the optical heterostructure phenomenon and twin structures first emerge during growth—closely matches that of the PL-enhanced inner zone in fully developed nanosheets (both $\approx 5 \mu\text{m}^2$; Figure R3.1, also see Figure 4d in the main text). Moreover, as the nanosheets grow laterally, the outer zone continuously expands while the inner zone remains unchanged, reflecting the fixed size of the transition zone (Supplementary Figure 20). This one-to-one spatial correspondence between the structural transition and the PL enhancement region provides compelling evidence that the OH phenomenon originates directly from the structural transition.

Figure R3.1 Statistical analysis of the areas of the transition and PL-enhanced inner zones.

Figure R3.2 Time-resolved PL (TRPL) spectra of the inner and outer zones, revealing a shorter lifetime and an additional fast radiative recombination pathway in the inner zone.

(2) Mechanistically, the enhanced emission in the inner zone stems from the structural transition. Time-resolved photoluminescence (TRPL) measurements reveal that the inner zone—corresponding to the structural transition region—exhibits a

markedly shorter PL lifetime but a significantly higher emission intensity than the outer zone (Figure R3.2, also see Figure 4e in the main text). Detailed fitting of the PL decay curves reveals an additional ultrafast radiative component (~ 0.15 ns) unique to the inner zone, which is absent in the outer region. This newly introduced fast radiative recombination pathway, triggered by the structural transition, accelerates carrier recombination and enhances the overall PL efficiency, thereby accounting for the pronounced emission enhancement localized within the inner zone. To determine whether the difference in PL intensity originates from variations in absorption, we have also measured the reflectance and transmittance spectra of both regions (as shown in Figure 1e of the revised main text), indicating negligible differences in optical absorption between the inner and outer zones.

Together, these results unambiguously demonstrate that the structural transition serves as the physical origin of the optical heterostructure, both in spatial correspondence and in carrier recombination dynamics.

2). Evidences for the existence of the structural transition.

(1) Clear contrasts in surface potential and phase are observed between the inner and outer zones of the nanosheet, as revealed by Kelvin probe force microscopy (KPFM) and electrostatic force microscopy (EFM) (Figure R3.3a, also see Figure 4a in the main text). These contrasts arise from variations in the local work function and electrostatic force gradient, respectively—both of which are highly sensitive to crystallinity differences induced by the structural transition^{18, 19}.

Figure R3.3 Optical microscope (OM), fluorescence (FL), Kelvin probe force microscopy (KPFM) and electrostatic force microscopy (EFM) images of a pristine thick PDBCz nanosheet (**a**, ~ 128 nm) and a pristine thin nanosheet (**b**, ~ 19 nm), respectively. After annealing, the OM and atomic force microscope (AFM) images

reveal significant structural degradation localized in the central region of the thick nanosheet, whereas the thin nanosheet remains intact.

(2) Since structural transitions typically render materials fragile and thermally unstable²⁰⁻²², we performed thermal annealing tests on PDBCz nanosheets. Remarkably, degradative damage occurred exclusively in nanosheets exhibiting the optical heterostructure phenomenon, and this damage was consistently confined to the inner zone, as revealed by optical and AFM images (Figure R3.3a, also see Figure 4a in the main text). In contrast, nanosheets without the OH phenomenon—whether artificially thinned, as-grown thin, or solution-processed—showed no discernible difference between the inner and outer zones after annealing (Figure R3.3b, also see Figure 4b in the main text and Supplementary Figures 11 and 16). These results demonstrate that the structural transition is present only in nanosheets displaying the OH phenomenon and is spatially confined to the PL-enhanced inner region.

(3) We conducted a series of thinning experiments to confirm the existence of the structural transition in the top layer. Thick nanosheets were gradually thinned by plasma etching. As the etching time increased, the nanosheet thickness decreased, and the luminescence contrast between the inner and outer zones progressively diminished (Supplementary Figure 13). Notably, when the thickness was reduced to a critical value of ~20 nm, the optical heterostructure phenomenon disappeared completely (Supplementary Figures 13-15). Similarly, as-grown nanosheets thinner than 20 nm exhibited no optical heterostructure behavior (Figure R3.3b, also see Figure 4b in the main text). Remarkably, both the artificially thinned and as-grown thin nanosheets remained intact after thermal annealing (Figure R3.3b, also see Fig. 4b and Supplementary Figure 16). Furthermore, the diffraction patterns of these thin nanosheets displayed only a single lattice orientation without any twin feature (inset of Figure 4c in the main text). These results confirm that both the structural transition and the OH phenomenon occur only in relatively thick nanosheets.

3). Origin of the structural transition.

During the island growth of the nanosheets, the top and bottom layers initially share the same lattice structure (m_1 , Figures 2c and 3b in the main text) at the early growth stage (stage I in Figure R3.4, also see Figure 3e in the main text). This uniformity arises

from strong molecule-substrate interactions (Figures 2g-h in the main text), which dominate in small-sized nanosheets^{23, 24}. As the nanosheet expands laterally, intermolecular interactions become increasingly dominant. When the nanosheet reaches a critical size (stage II in Figure R3.4, also see Figure 3e in the main text), the lattice of the top layer undergoes a transition to the m_2 orientation—maintaining the same crystal framework but rotating relative to m_1 —while the bottom layer remains locked in the m_1 orientation due to its intimate contact with the substrate. This leads to the emergence of a twin structure comprising $m_1 + m_2$ domains (Figures 2b and 3b). The structural transition is enabled by the siliceous substrate, which forms strong interfacial interactions with the molecules, possibly through Si-Br covalent bonding²⁵ (Figures 2g-h in the main text). Following the structural transition, the nanosheets undergo further lateral growth, during which the bottom layer retains the m_1 lattice while the newly formed top layer preserves the m_2 lattice (stage III in Figure R3.4, also see Figure 3e in the main text).

Figure R3.4 Schematic diagram of the optical heterostructure mechanism of a PDBCz nanosheet with three important stages of growth process.

Comment 2:

The authors suggest that the formation of the heterostructure is strongly influenced by interactions between the PDBCz nanosheets and substrates, yet no experiments were conducted to investigate these interfacial interactions.

Reply to Comment 2:

We sincerely thank the reviewer for this valuable comment. We fully agree that direct characterization of molecule-substrate interfacial interactions is extremely challenging, especially for molecularly thin organic nanosheets, where the interface is buried and conventional surface-sensitive techniques (such as XPS or STM) are difficult to apply without disturbing the crystal structure. To address this issue, we performed a

systematic series of controlled experiments to indirectly yet convincingly probe the interfacial effect, as summarized in Table R3.1 as follows.

First, PDBCz nanosheets grown by physical vapor deposition on siliceous substrates (SiO_2/Si , Si, and quartz) consistently exhibited the optical heterostructure phenomenon and the twin structure (Figure 1c in the main text and Supplementary Figure 7). In contrast, PDBCz nanosheets grown on non-siliceous substrates (BN, mica, and sapphire) and solution-processed nanosheets (where molecule-substrate interactions are negligible) displayed uniform photoluminescence without any optical heterostructure feature (Figures 2d-e in the main text and Supplementary Figure 6).

Second, to further isolate the chemical contribution of substrate elements, we synthesized and deposited structurally similar carbazole derivatives lacking bromine atoms, namely 1,4-di(9H-carbazol-9-yl)benzene (PDCz) and 1,3-bis(N-carbazolyl)benzene (mCP), on the SiO_2/Si substrates. Neither of these bromine-free molecules exhibited the optical heterostructure phenomenon. These results indicate that the interfacial interaction between Si and Br plays a crucial role in triggering the structural transition responsible for the optical heterostructure (Supplementary Figure 8).

Third, first-principles calculations further support this interpretation. For the PDBCz- SiO_2 system, the adsorption energy (-0.764 eV) is lower than both its intermolecular interaction energy (-0.723 eV) and that of the PDBCz-BN system (close to 0 eV), while the PDCz- SiO_2 system (-0.443 eV) shows a higher adsorption energy than its intermolecular interaction energy (-1.071 eV) (Figure 2h in the main text). This contrast reveals that the PDBCz- SiO_2 interface possesses markedly stronger binding strength, primarily induced by Si-Br bonding. Consistently, charge-density analysis (Supplementary Figure 9) clearly shows electron density accumulation between Si and Br atoms, confirming the formation of covalent Si-Br bonds at the interface.

Together, these results provide compelling evidence that the optical heterostructure phenomenon originates from strong interfacial coupling between PDBCz and siliceous substrates via Si-Br bonding, which modulates the balance between interfacial and intermolecular interactions during crystal growth. Although direct measurement of this buried interface is experimentally challenging, the combined comparative experiments and theoretical calculations unambiguously reveal the decisive role of interfacial interactions in the formation of the heterostructure.

Table R3.1 Summary of controlled experiments on substrate and molecular dependence of the optical heterostructure phenomenon.

Growth Method	Substrate Type	Molecular Structure	Presence of Si Element	Presence of Br Element	Observation of optical heterostructure	Adsorption energy (eV)	Interpretation
PVD	SiO ₂ /Si	PDBCz	✓	✓	Yes	-0.764	Lower than its intermolecular interaction energy (-0.723 eV). Strong Si-Br interfacial interaction induces twin structure and optical heterostructure.
PVD	Quartz	PDBCz	✓	✓	Yes	---	Strong Si-Br interfacial interaction induces twin structure and optical heterostructure.
PVD	BN	PDBCz	✗	✓	No	0	Weak van der Waals interaction, absence of Si-Br bonding.
PVD	Mica/Sapphire	PDBCz	✗	✓	No	---	Non-siliceous substrates, weak interfacial coupling.
PVD	SiO ₂ /Si	PDCz/mCP	✓	✗	No	-0.443	Higher than its intermolecular interaction energy (-1.071 eV). Lack of Br prevents Si-Br bond formation.
Solution growth	SiO ₂ /Si	PDBCz	✓	✓	No	---	Weak molecule-substrate interaction under solution growth.

Comment 3:

Did the authors examine the emission properties from different sides of the bulk crystals? Is there any variation in emission intensity?

Reply to Comment 3:

We thank the reviewer for raising this important point about facet-dependent emission, a key property for bulk crystals. In our work, we intentionally employ a PVD method based on strong molecule-substrate interactions and island growth to form oriented nanosheets, as these are directly relevant for the optical heterostructures we target. While this approach does not yield the bulk crystals necessary for the suggested experiment, it allows us to probe a different but equally critical aspect: in-plane uniformity. Critically, we would like to emphasize that the observed contrast in emission between the inner and outer zones is not a consequence of differing crystallographic orientations. We confirm that all PDBCz nanosheets exhibit identical crystallographic orientation along the *b*-axis, as demonstrated by both selected-area electron diffraction (SAED) and X-ray diffraction (XRD) analyses.

Specifically, the SAED patterns collected from multiple zones of the same nanosheet (Figure 2a in the main text and Supplementary Figures 4-5) show consistent diffraction spots corresponding to the (020) family planes, indicating that the entire nanosheet—including both the inner and outer regions—shares the same in-plane orientation. The XRD data (Supplementary Figure 1) further show sharp and single (020) diffraction peaks, confirming the preferred orientation of all nanosheets along the *b*-axis. Therefore, the observed PL contrast between the inner and outer regions cannot originate from different exposed crystal facets, as the crystal orientation remains identical across the nanosheet.

Furthermore, theoretical calculations of the relative growth rates and surface free energies of the major crystal facets reveal that the (020) surface—which corresponds to the *b*-axis orientation—possesses the lowest surface energy and slowest growth rate (Figure 3c in the main text). This theoretical result is fully consistent with the experimental observation that PDBCz nanosheets preferentially grow with the (020) facet exposed parallel to the substrate.

Together, these results unequivocally demonstrate that all nanosheets are oriented along the *b*-axis, and the PL enhancement in the inner zone does not stem from different crystal faces. Instead, it arises from the out-of-plane twin structural transition induced

by the interplay between molecule-substrate and intermolecular interactions, as described in the main text.

Comment 4:

Many key conclusions in the manuscript are speculative and lack direct evidence, which is not scientifically rigorous. The authors should perform more detailed mechanistic studies.

Reply to Comment 4:

We sincerely thank the reviewer for this critical comment, which has helped us significantly strengthen the mechanistic discussion in our manuscript. We agree that scientific rigor is paramount and have undertaken a comprehensive effort to move from the initial observation of a novel phenomenon toward a more substantiated mechanistic understanding.

The core of our revised argument is that the observed optical heterostructure—the brighter emission from the inner zone of the nanosheet—exists precisely because a solid-state structural transition to a twin crystal occurs locally in the top layer of the inner zone, and this transition locally enhances the photoluminescence quantum yield.

This unified model is supported by the following key lines of evidence, which we have elaborated in the revised text:

A. Evidence for a Localized Twin-Structure Transition:

1. **Direct Structural Evolution:** Structural characterization of nanosheets at different growth stages reveals a clear evolution: the selected area electron diffraction (SAED) pattern transitions from a single-crystal (m_1) pattern to a twin-crystal ($m_1 + m_2$) pattern (Figure 3b).

2. **Spatial Localization to the Center:** The inner zone exhibits distinct behavior under thermal stress (showing damage) and in electrical potential (KPFM/EFM), confirming its unique structural identity compared to the outer zone without transition zone (new data in Figures 4a-b).

3. **Confinement to the Top Layer:** Thin nanosheets (<20 nm) exhibit only the single-crystal pattern and show no differences between inner and outer zones in all

characterizations, proving that the structural transition is confined to the top layers of the nanosheet (new data in Figure 4b and Supplementary Figures 13-15).

4. **Driving Force Identified:** The strong molecule-substrate interaction, primarily via Si-Br bonding, is the key driver. This is confirmed by control experiments: nanosheets grown from solution method, on BN substrates, or using Br-free analogous molecules all fail to form the twin structure (Figures 2d-h including some new data).

5. **Energetic Plausibility of the Growth Mode:** Theoretical calculations confirm that the observed island-growth mode and the resulting coherent lattice between inner and outer zones are energetically favorable (new data in Figure 3c-d).

B. Evidence Correlating the Twin Transition with the Optical Heterostructure:

1. **Temporal Co-occurrence:** The twin structure and the optical heterostructure emerge simultaneously during the dynamic growth process (Figure 3b).

2. **Universal Co-existence:** The two phenomena consistently appear together across a wide range of conditions, including different molecular designs, growth methods, substrates, and thicknesses, as shown in Figures 2d-e and Supplementary Figures 7-8 with some new data.

3. **Universal Absence in their Absence:** Thin nanosheets (< 20 nm) that do not undergo the structural transition also show no sign of the OH, demonstrating that one is a prerequisite for the other (new data in Figure 4b).

4. **Spatial Correlation:** The spatial extent of the structurally transformed region aligns almost perfectly with the bright inner zone region of the optical heterostructure (Figure 4d).

5. **Mechanism for Enhanced Emission:** Time-resolved PL (new data in Figure 4e) reveals a faster and more complex decay in the inner zone, while absorbance remains unchanged (new data in Figure 1e), directly pointing to an enhanced radiative efficiency in the transition region.

6. **Quantitative Model Validation:** The proposed model—where only the top layer of the inner zone has enhanced intensity (I_i) and all other regions have a baseline intensity (I_0)—perfectly fits the PL statistical data from a large number of nanosheets of different thicknesses (Figure 4c with new description in Supplementary Note V).

We acknowledge the profound challenge of obtaining direct, molecular-level

evidence of dynamic structural transitions in fragile organic micro-nano systems, where real-time tracking of transient phases remains exceptionally difficult. Over the past six years, we have exhaustively employed all accessible characterization techniques to construct the multi-faceted, corroborating evidence presented here.

We are convinced that the discovery of an optically distinct heterostructure arising from a solid-state transition in an organic system opens a new and fertile platform for exploration. To actively encourage this, we have explicitly outlined the open questions and future possibilities in the revised manuscript. By framing our work in this way, we hope to invite the broader community to join us in uncovering deeper mechanistic insights and more compelling evidence. We believe this phenomenon will stimulate further thinking and highlight that organic micro-nano crystals host a wealth of unexplored science related to structural dynamics and property control. We hope the reviewer finds our strengthened evidence and clarified discussion persuasive and agrees that our work provides a valuable and intriguing foundation for future studies.

References

1. Leyre, S., et al. Absolute determination of photoluminescence quantum efficiency using an integrating sphere setup. *Rev. Sci. Instrum.* **85**, 12 (2014)
2. Würth, C., et al. Evaluation of a commercial integrating sphere setup for the determination of absolute photoluminescence quantum yields of dilute dye solutions. *Appl. spectrosc.* **64**, 733-741 (2010)
3. Deng, S., et al. Long-range exciton transport and slow annihilation in two-dimensional hybrid perovskites. *Nat. Commun.* **11**, 664 (2020)
4. Shi, H., et al. Highly efficient ultralong organic phosphorescence through intramolecular-space heavy-atom effect. *J. Phys. Chem. Lett.* **10**, 595-600 (2019)
5. Kautny, P., Schwartz, T., Stöger, B. & Fröhlich, J. An unusual case of OD-allotwinning: 9,9'-(2,5-dibromo-1,4-phenylene)bis[9H-carbazole]. *Acta Crystall. B* **73**, 65-73 (2017)
6. Akselrod, G. M., et al. Visualization of exciton transport in ordered and disordered molecular solids. *Nat. Commun.* **5**, 3646 (2014)
7. Tempelaar, R., Jansen, T. L. C. & Knoester, J. Exciton-exciton annihilation is coherently suppressed in H-aggregates, but not in J-aggregates. *J. Phys. Chem. Lett.* **8**, 6113-6117 (2017)
8. Kumar, S., et al. Exciton annihilation in molecular aggregates suppressed through quantum interference. *Nat. Chem.* **15**, 1118-1126 (2023)
9. Liu, K., et al. Tunable microstructures of ultralong organic phosphorescence materials. *Chem. Commun.* **57**, 7276-7279 (2021)
10. Docherty, R., Clydesdale, G., Roberts, K. & Bennema, P. Application of Bravais-Friedel-Donnay-Harker, attachment energy and Ising models to predicting and understanding the morphology of molecular crystals. *J. Phys. D: Appl. Phys.* **24**, 89 (1991)
11. Berkovitch-Yellin, Z. Toward an ab initio derivation of crystal morphology. *J. Am. Chem. Soc.* **107**, 8239-8253 (1985)
12. The Collected Works of J. Willard Gibbs. *Nature* **124**, 119-120 (1929)
13. Li, R., et al. Gibbs-Curie-Wulff theorem in organic materials: a case study on the relationship between surface energy and crystal growth. *Adv. Mater.* **28**, 1697-1702 (2016)
14. Xu, W., et al. Asymmetric charge carrier transfer and transport in planar lead halide

- perovskite solar cells. *Cell Reports Physical Science* **3**, 5 (2022)
15. Tanabe, I., et al. Electronic excitation spectra of organic semiconductor/ionic liquid interface by electrochemical attenuated total reflectance spectroscopy. *Commun. Chem.* **4**, 88 (2021)
 16. de Souza, G. F., et al. Probing the cw-laser-induced fluorescence enhancement in CsPbBr₃ nanocrystal thin films: An interplay between photo and thermal activation. *ACS Appl. Mater. Interfaces* **16**, 34303-34312 (2024)
 17. Moral, M., Son, W. J., Sancho-Garcia, J. C., Olivier, Y. & Muccioli, L. Cost-effective force field tailored for solid-phase simulations of OLED materials. *J Chem. Theory Comput.* **11**, 3383-3392 (2015)
 18. Bergmann, V. W., et al. Real-space observation of unbalanced charge distribution inside a perovskite-sensitized solar cell. *Nat. Commun.* **5**, 5001 (2014)
 19. Li, T. & Zeng, K. Probing of local multifield coupling phenomena of advanced materials by scanning probe microscopy techniques. *Adv. Mater.* **30**, e1803064 (2018)
 20. Ran, N. A., et al. Impact of interfacial molecular orientation on radiative recombination and charge generation efficiency. *Nat. Commun.* **8**, 79 (2017)
 21. Cuthriell, S. A., Malliakas, C. D., Kanatzidis, M. G. & Schaller, R. D. Cyclic versus linear alkylammonium cations: preventing phase transitions at operational temperatures in 2D perovskites. *J. Am. Chem. Soc.* **145**, 11710-11716 (2023)
 22. Namakian, R., Garzon, M. A., Tu, Q., Erdemir, A. & Gao, W. Temperature-induced phase transition in 2D alkylammonium lead halide perovskites: A molecular dynamics study. *ACS Nano* **18**, 22926-22937 (2024)
 23. Zhang, Y., et al. Probing carrier transport and structure-property relationship of highly ordered organic semiconductors at the two-dimensional limit. *Phys. Rev. Lett.* **116**, 016602 (2016)
 24. Xu, C., et al. A general method for growing two-dimensional crystals of organic semiconductors by "solution epitaxy". *Angew. Chem. Int. Ed.* **55**, 9519-9523 (2016)
 25. Turunen, L. & Erdélyi, M. Halogen bonds of halonium ions. *Chem. Soc. Rev.* **49**, 2688-2700 (2020)